# Dedicated photoreceptor pathways in *Drosophila* larvae mediate navigation by processing either spatial or temporal cues

Tim-Henning Humberg [1], Pascal Bruegger[1], Bruno Afonso[2], Marta Zlatic[2,3], James W. Truman[2], Marc Gershow [4], Aravinthan Samuel[5] & Simon G. Sprecher [1]

To integrate changing environmental cues with high spatial and temporal resolution is critical for animals to orient themselves. *Drosophila* larvae show an effective motor program to navigate away from light sources. How the larval visual circuit processes light stimuli to control navigational decision remains unknown. The larval visual system is composed of two sensory input channels, Rhodopsin5 (Rh5) and Rhodopsin6 (Rh6) expressing photoreceptors (PRs). We here characterize how spatial and temporal information are used to control navigation. Rh6-PRs are required to perceive temporal changes of light intensity during head casts, while Rh5-PRs are required to control behaviors that allow navigation in response to spatial cues. We characterize how distinct behaviors are modulated and identify parallel acting and converging features of the visual circuit. Functional features of the larval visual circuit highlight the principle of how early in a sensory circuit distinct behaviors may be computed by partly overlapping sensory pathways.

[1] Department of Biology, University of Fribourg, 1700 Fribourg, Switzerland. [2] Janelia Research Campus, Howard Hughes Medical Institute, Ashburn 20147 VA, USA. [3] Department of Zoology, University of Cambridge, CB2 3EJ Cambridge, UK. [4] Department of Physics and Center for Neural Science, New York University, New York 10003 NY USA. [5] Department of Physics and Center for Brain Science, Harvard University, Cambridge 02138 MA, USA. Correspondence and requests for materials should be addressed to S.G.S. (email: simon.sprecher@unifr.ch)

Goal-directed navigation is an important biological achievement allowing animals to continuously evaluate changes in their environment to base behavioral decisions on these inputs. The mechanisms of how sensory neurons and the underlying neural networks control navigational decisions remain still largely unknown.

For efficient navigation, an animal may detect stimulus changes in time, space or both[1,2]. A single sensory receptor is sufficient for taxis, if the animal compares the current stimulus with a past stimulus strength (klinotaxis)[1,2]. Intensity changes in space, for instance along a gradient, can be either detected by instantaneous comparisons of pairs of sensory neurons or sense organs located at the different sites of the body (tropotaxis), or by moving a single sensory neuron in space, thus translating a spatial intensity change into a temporal intensity change (klinotaxis)[1,2].

Larvae of the fruit fly *Drosophila melanogaster* navigate efficiently in response to stimuli of different sensory modalities including temperature (thermotaxis), odorants (chemotaxis) or light (phototaxis)[3–9]. How integration of spatial and temporal information may contribute to effective navigation of larvae remains largely unknown.

Larval locomotion consists of two motor programs, namely phases of forward movements (called runs), which are spaced by reorientation events (called turns)[10] (Fig. 1a). During a turn the animal probes the environment by sweeping its head to one side, a process allowing a translation of spatial intensity differences into temporal information[4–7] (Fig. 1a). A head sweep toward an unfavorable direction is more likely followed by another head sweep toward the preferred direction, than a first head sweep, which is already pointing to a preferred direction[4–7] (Fig. 1a). The new run direction is determined by the final head sweep acceptance[4–7] (Fig. 1a). Thus, the number and direction of head sweeps may be modulated to make a turn toward the preferred direction. Larvae may also change the direction during runs with respect to

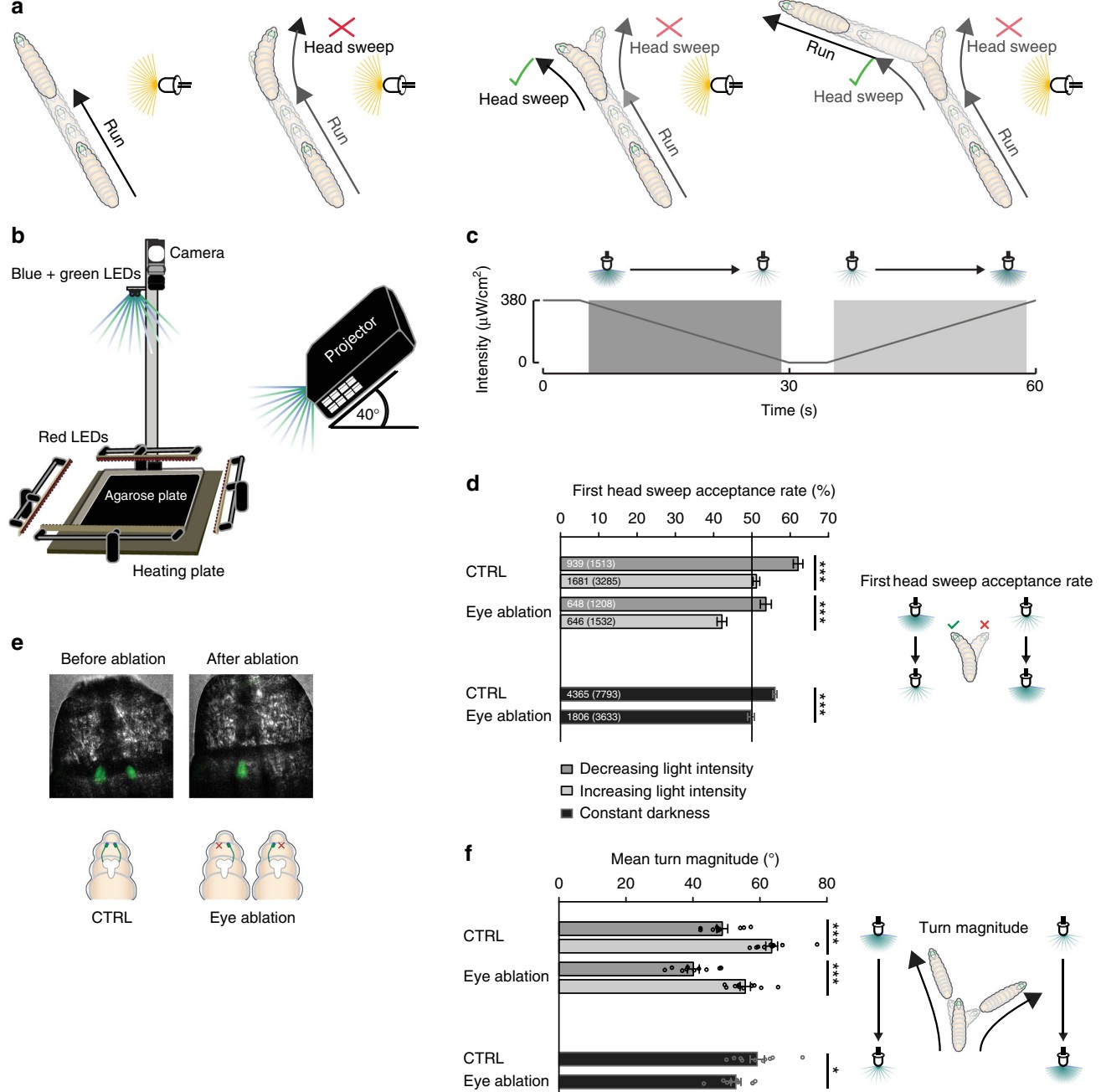

a stimulus source by curving the run toward the preferred direction[7].

Larvae prefer shaded over illuminated areas. Presumably, this dark preference is advantageous, because in light larvae are more exposed to predators, DNA-damaging ultraviolet light or risk of desiccation. Visual cues are perceived by specialized photoreceptor neurons (PRs) of their paired eyes, which closely associated with the head skeleton[11]. Light perception by the PRs is essential for light navigation[6,9]. Visually guided navigation is directly implemented in a range of complex behaviors including formation of associative memories, circadian rhythmic activity and the ability to respond to movement of other larvae[12–19]. During phototaxis we recently found that larvae bias the turn direction away from a light source and that they perform bigger turns when they are heading toward a light source compared with heading away from it[6,9]. Both navigational strategies are proposed to be mediated by temporal processing of light information during head casts[6]. In a lightscape of spatially varying light intensities, a head sweep will lead to a change in light intensity on the larval eyes over time. Larvae could be able to compare the present with the past light intensity. For example, a detected decrease in light intensity (between these two time points) during a head sweep should result rather in accepting than rejecting this head sweep and as consequence turning toward the darker direction. If larvae may also use spatial cues for navigation remains unknown. Electrophysiological recordings in the blowfly larva show that by shining light only on one side the strength of spiking of larval PRs greatly differs between the light-exposed and the less light-exposed eyes, suggesting that a bilateral comparison of visual input may be employed by larvae[1,20].

Each larval eye houses about 12 PRs, which can be subdivided into two types defined by the *Rhodopsin* gene that they express. Four PRs express the blue-tuned *Rhodopsin5* (*Rh5*), the remaining PRs express green-tuned *Rhodopsin6* (*Rh6*)[21–26]. Several studies show that Rh5-PRs are essential for light avoidance, while Rh6-PRs appear dispensable[6,13,27]. However, we recently found that in natural lighting situations both PR subtypes are necessary for navigation, showing that Rh6-PRs do have a direct role for visual guided navigation[9].

PRs project their axons into the larval brain connecting to their target neurons in the larval optic neuropil (LON)[13]. PRs target neurons can be divided into two types: visual local neurons and visual projection neurons. Visual local neurons do not extend neurites beyond the LON and therefore function uniquely to process visual information within the LON, while visual projection neurons extend to defined domains of the central brain and therefore transmit visual input toward higher brain centers[28]. Interestingly, Rh6-PRs predominantly synapse onto visual local neurons, while Rh5-PRs predominantly connect to visual projection neurons[28]. Visual local neurons in turn synapse onto most visual projection neurons suggesting a modulatory role for the Rh6-PRs to visual local neurons pathway, in agreement with a nonessential role for Rh6-PRs in phototaxis under laboratory conditions[6,9,28]. How Rh6-PRs function and how the different types of visual interneurons are involved in visually guided behaviors remains unknown.

We here decipher how bilateral information, two PRs types and distinct sets of visual interneurons contribute to navigational decision making. By combining a computer based tracking system with eye ablation experiments, we show that binocularity in larvae is critical for phototaxis, allowing animals to integrate temporal and spatial light information. Interestingly, the two input channels of the larval visual system mediate phototaxis by task sharing of temporal and spatial information processing. Rh6-PRs are essential for the perception of temporal light information during head casts and thus contribute to better navigation. By following the information flow into the visual circuit, we further found that a specific set of direct PR targets the so-called "optic lobe pioneer neurons" (OLPs) are required for a navigational behavior that depends on temporal light cues, comparable to the Rh6-PRs. Moreover, while Rh5-PRs are essential for integration of spatial light information, Rh6-PRs are dispensable. The ability of spatial information processing by comparison of the output of two sensory organs and the dedication of the two PR types for spatial and temporal coding provides an efficient system for navigation.

## Results

**Larvae use temporal and spatial cues for phototaxis**. Navigational strategies underlying phototaxis in *Drosophila* larvae can be mediated by temporally comparing light cues, however if spatial information may be used by the animal for navigation remains unknown[6]. The larval eye lacks accessory cells or an ommatidial organization comparable to the adult compound eye. Within one eye there are no apparent anatomical features that would suggest for spatial light perception. However, the presence of a pair of eyes in principle allows binocular comparison and thus spatial light perception. We therefore compared wildtype animals with animals that had one eye ablated for their ability to navigate. In

**Fig. 1** Larval navigation. **a** Schematics demonstrate basic principles of larval navigation. A larva stops its run and probe the environment by head sweeping. A rejected head sweep is followed by another head sweep. An accepted head sweep is followed by a run in this new direction. **b** Schematic drawing of the behavioral setup. Approximately 30 larvae moved freely on an agarose plate. The temperature of the agarose plate was controlled by a heating plate. The testing plate was illuminated by red LEDs. A camera recorded larval behavior. Larvae were stimulated with a projector from one side or with LEDs from top. **c** In experiments where a temporal changing light stimulus was presented, the light intensity was decreasing linearly from 380 to 0 $\mu W/cm^2$ for 25.5 s and increasing linearly from 0 to 380 $\mu W/cm^2$ with the same speed. These phases were spaced by 4.5 s constant high or low light intensity. The phases of constant light intensity ±1 s were not taken into consideration for analysis. We analyzed behavior occurring in time phase of linear intensity decrease (dark gray) and linear intensity increase (light gray), respectively. **d** More head sweeps are accepted during the phases of light intensity decrease compared with phases of light intensity increase. One eye is sufficient for this behavioral bias. Fisher's exact test with $n$ = number of first head sweeps: CTRL: $n$ = 4798, $p$ = 1.7 × 10$^{-12}$; eye ablation: $n$ = 2740, $p$ = 2.8 × 10$^{-9}$. In darkness one-eyed larvae accepted less head sweeps compared to control animals. Fisher's exact test with $n$ = number of first head sweeps: CTRL (darkness) vs eye ablation (darkness): $n$ = 11426, $p$ = 3.6 × 10$^{-10}$. **e** We created unilateral seeing larvae by the use of an ablation laser. Bolwig organs are marked in green by expression of green fluorescent protein. **f** During phases of light intensity increase the magnitude of turns was greater than the phase of light intensity decrease. One eye was sufficient for this navigational parameter. Two sample $t$-test with $n$ = numbers of experiments: CTRL: $n$ = 10, $p$ = 3.4 × 10$^{-6}$; eye ablation: $n$ = 10, $p$ = 3.4 × 10$^{-6}$. One-eyed animals made smaller turns in comparison with control larvae in constant darkness. Two sample $t$-test with $n$ = numbers of experiments: CTRL (darkness) vs eye ablation (darkness): $n$ = 20, $p$ = 0.024. **d**, **f** Benjamini Hochberg procedure was used to adjust $p$-values for multiple testing. *$p$ < 0.05, ***$p$ < 0.001. Exact $F$-values, $t$-values and degrees of freedom can be found in Supplementary Table 1. The data show mean and error bars show SEM. SEM of binary choice data was calculated with the formula (1). **d** The first number is the number of accepted first head sweeps and the number in brackets is the total number of first head sweeps for the two phases, respectively. **f** Circles indicate mean of individual experiments

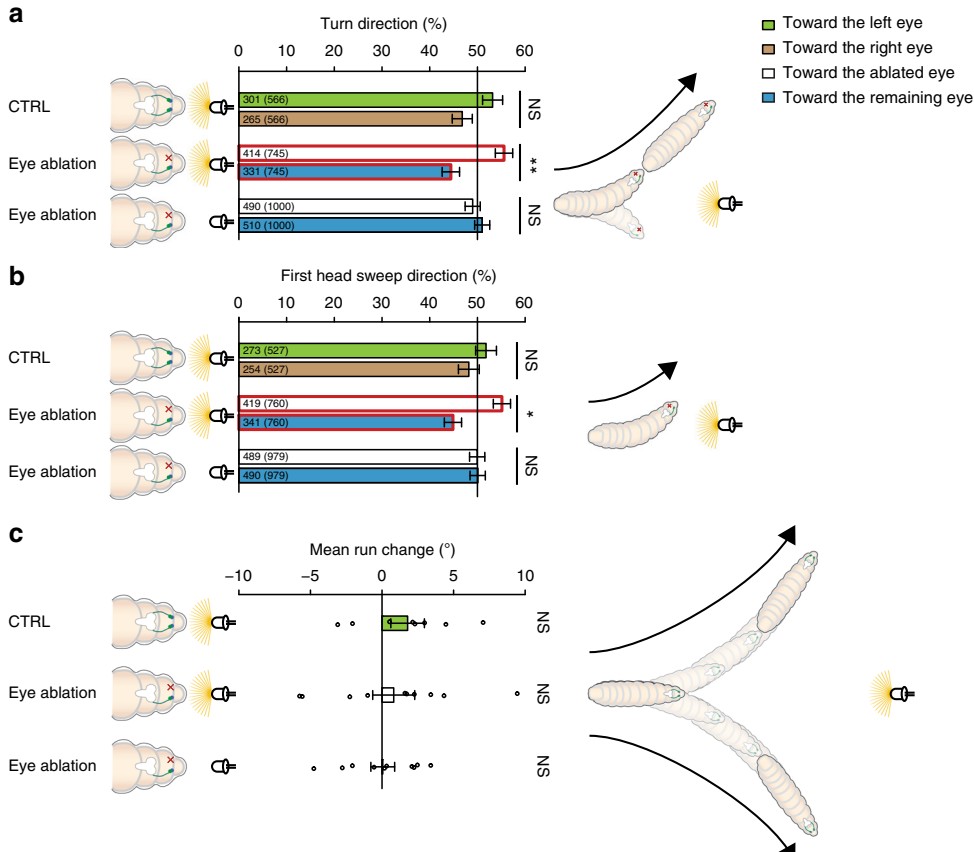

**Fig. 2** Unilateral sensing larvae bias their first head sweep direction and their turn direction toward the side of their blinded eye, when they are heading toward a light source. **a** Bilateral sensing control larvae do not bias their turn direction when they are facing a light source. Unilateral sensing animals bias their turn direction toward the blinded eye when heading toward a light source. These larvae do not bias their turn direction in absence of a light stimulus. Exact binomial test with $n$ = numbers of turns: CTRL: $n$ = 566, $p$ = 0.2118; eye ablation light on: $n$ = 745, $p$ = 0.0079; eye ablation light off: $n$ = 1000, $p$ = 0.548. **b** Bilateral sensing larvae do not bias the first head sweep direction when facing toward a light source. Unilateral sensing animals bias the first head sweep toward the side of the blinded eye, when they are facing toward a light source. Exact binomial test with $n$ = numbers of first head sweeps: CTRL: $n$ = 527, $p$ = 0.6495; eye ablation light on: $n$ = 760, $p$ = 0.0156; eye ablation light off: $n$ = 979, $p$ = 1. **c** Neither bi- nor unilateral sensing larvae bias their steering within runs when heading toward a light source. One sample $t$-test with $n$ = number of experiments: CTRL: $n$ = 8, $p$ = 0.5139; eye ablation light on: $n$ = 10, $p$ = 0.8865; eye ablation light off: $n$ = 10, $p$ = 0.9516. **a–c** Benjamini Hochberg procedure was used to adjust $p$-values for multiple testing. *$p$ < 0.05, **$p$ < 0.01, $n$ = not significant. Exact $F$-values, $t$-values and degrees of freedom can be found in Supplementary Table 1. The data show mean and error bars show SEM. **a** The first number is the number of turns in the indicated direction (please see color code) and the number in brackets is the total number of turns in both directions. **b** The first number is the number of first head sweeps directed to the indicated direction and the number in brackets is the total number of first head sweeps in both directions. **c** Circles represent means of individual experiments

our study, we used a phototaxis assay as previously described[6] (Fig. 1b). Briefly, per experiment 30 larvae were moving freely on an agarose plate for 11 min. Larvae were stimulated with different lighting schemes, either with a directional light source (projector from one side) or with a temporally varying light source (light-emitting diodes (LEDs) from above) (Fig. 1b).

A larva which swings its head toward a light source is likely to experience an increase of light stimulation over time, whereas a head sweep away from the light should result in a temporal decrease of light stimulation. In a first assay, we exposed animals to temporally varying, but spatially uniform lightscape to test uniquely for temporal changes in light stimulation (Fig. 1c). We used a temporal ramp during which 23.5 s light intensity decreases (dark gray) or increases linearly (light gray). The total amount of light intensity presented to the animal during both periods was identical (Fig. 1c). We analyzed the first head sweep acceptance rate by calculating the percentage of accepted first head sweeps (these head sweeps are terminating the turn and are

followed by a run) from all first head sweeps, which occurred during both periods, respectively. We observed that the first head sweep acceptance rate was much lower in phases of increasing light intensity compared to phases when the light intensity decreases (Fig. 1d). During light increase it was as likely to reject a head sweep than accepting it, however during light decrease significantly more head sweeps were accepted than rejected (Fig. 1d). In line with previous results, this finding suggests that the decision to accept or reject a head sweep is based on temporal processing of light intensity during the head sweep[29] (Fig. 1d). In principle one eye could be sufficient to detect temporal changes in light intensity. We therefore performed experiments with one-eyed animals. Unilateral sensing larvae were created using a laser ablation system by specifically ablating either the left or right eye and each group was tested separately (Fig. 1e). Unilateral sensing animals also biased the first head sweep acceptance rate (Fig. 1d). In the phase of intensity increase more first head sweeps were rejected than accepted,

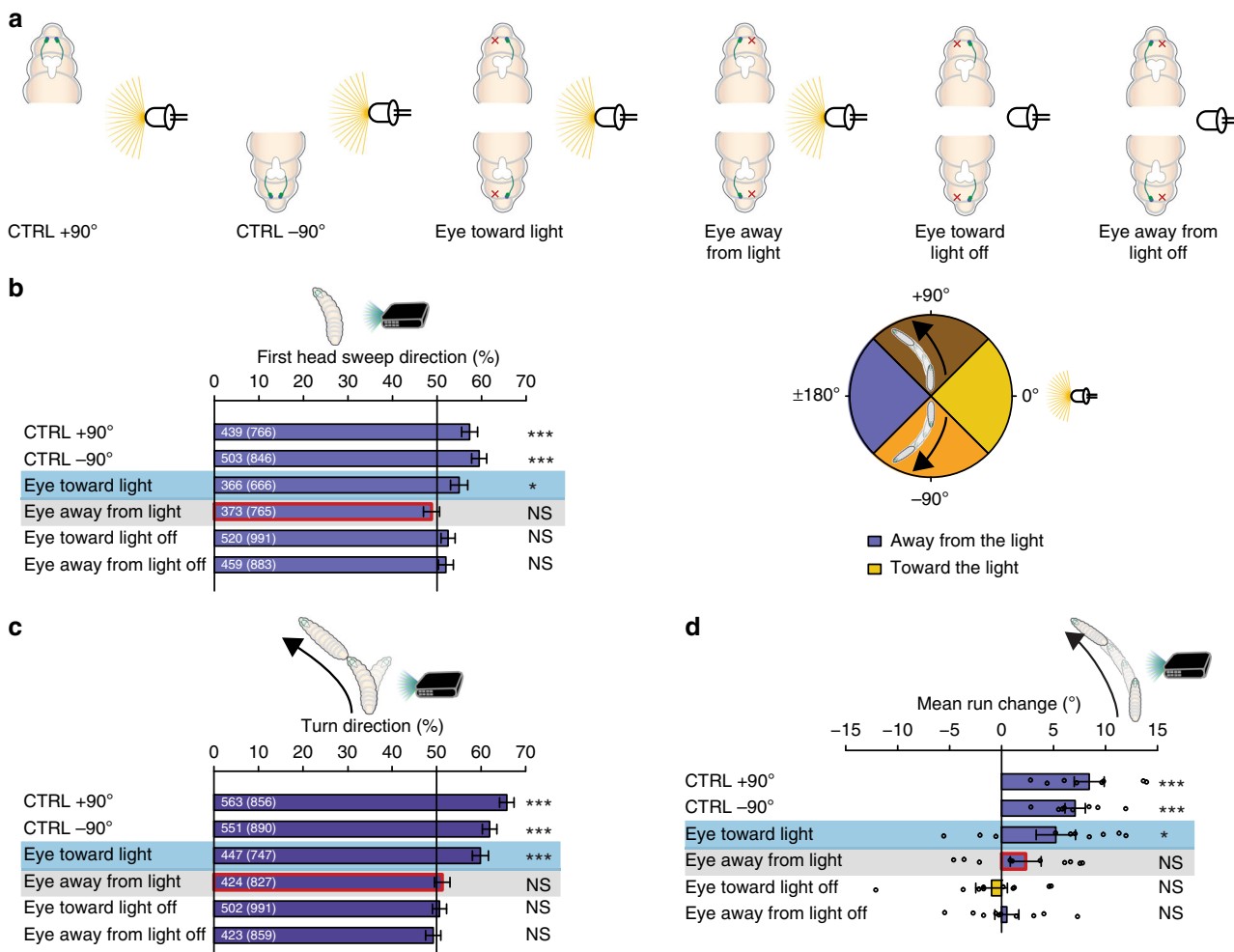

**Fig. 3** *Drosophila* larvae use navigational strategies based on spatial information integration to avoid a directional light source. **a** In dependence of heading direction is the light source on the ipsi- or contralateral side of the animal with respect to the functional eye of unilateral sensing larvae. **b** Heading to +90° (brown) larvae should bias their movements to the left, in order to avoid the light source. Larvae, which are orientated to −90° (orange), should bias their movements to the right to avoid the light source. Bilateral sensing animals bias their first head sweep direction away from the light source, independently of their initial heading direction. Larvae with their functional eye on the body side facing the light source bias their first head sweep direction, but not when the functional eye is on the side facing away from the light source. These animals do not bias their first head sweep to any direction in constant darkness. CTRL +90°: $n = 766$, $p = 1.8 \times 10^{-4}$; CTRL −90°: $n = 846$, $p = 2.5 \times 10^{-7}$; eye toward light: $n = 666$, $p = 0.0234$; eye away from light: $n = 765$, $p = 0.5152$; eye toward light off: $n = 991$, $p = 0.1910$; eye away from light off: $n = 883$, $p = 0.303$. **c** Being perpendicular to a light source, bilateral sensing animals bias their turn direction away from the light. Unilateral sensing larvae need the remaining eye on the body side toward the light, in order to bias the turn direction. CTRL +90°: $n = 856$, $p = 1.3 \times 10^{-15}$; CTRL −90°: $n = 890$, $p = 3.6 \times 10^{-12}$; eye toward light: $n = 747$, $p = 1.7 \times 10^{-12}$; eye away from light: $n = 827$, $p = 0.7031$; eye toward light off: $n = 991$, $p = 0.7031$; eye away from light off: $n = 859$, $p = 0.7031$. **d** Larvae steer within runs away from the light source. Unilateral sensing larvae need the functional eye facing toward the light source to bias this steering in runs. CTRL + 90°: $n = 8$, $p = 5.2 \times 10^{-4}$; CTRL −90°: $n = 8$, $p = 5.2 \times 10^{-4}$; eye toward light: $n = 10$, $p = 0.0442$; eye away from light: $n = 10$, $p = 0.2292$; eye toward light off: $n = 10$, $p = 0.6494$; eye away from light off: $n = 10$, $p = 0.673$. The data are represented as mean ± SEM. For statistical analysis the exact binomial test (**b**, **c**) and the one sample *t*-test (**d**) was applied. Benjamini Hochberg procedure was used to adjust *p*-values for multiple testing. Exact *F*-values, *t*-values and degrees of freedom can be found in Supplementary Table 1. **b** The first number gives the number of first head sweeps directed to the indicated direction (please see color code) and the number in brackets is the total number of first head sweeps in both directions. **c** The first number indicates the number of turns in the indicated direction and the number in brackets is the total number of turns in both directions. **d** Circles show means of individual experiments. *$p < 0.05$, ***$p < 0.001$, n = not significant

supporting the notion that one eye is sufficient for temporal integration during head sweeps. The overall first head sweep acceptance rates are lower in unilateral sensing larvae compared to binocular sensing animals. This decrease is likely a side effect of eye ablation per se, as also in constant darkness one-eyed animals have lower first head sweep acceptance rates compared to binocular sensing control larvae under the same conditions (Fig. 1d).

Another second navigational strategy is that larvae bias the turn size. The turn size was defined as the angle between larval heading before and after the turning event. The turn size was calculated for turns occurring in the phase of light intensity increase and decrease separately. Larvae made larger turns in the phase of increasing light intensity than in the phase of intensity decrease, in agreement with previous findings[6] (Fig. 1f). We found that unilateral sensing larvae also bias the size of their turns

in relation to the light intensity change (Fig. 1f), supporting that one eye is sufficient for temporal processing for this strategy. Similarly as for first head sweep acceptance we observed that the eye ablation has a general effect on the turn size, as one-eyed larvae make smaller turns in absence of a light stimulus in comparison to control animals (Fig. 1f). Both behavioral biases, turn magnitude and first head sweep acceptance rate can be elicited by temporal light gradients with different steepness (Supplementary Figure 1).

Beside taxis (directional bias of movements in relation to stimulus direction) larvae could also use kinesis (no directional bias of movements) to avoid light[6,30]. Larvae can change their turn rate related to light intensity increase or decrease, a behavioral decision that may be explained with kinesis[30]. Indeed, wildtype larvae turn less often when they sense a light intensity decrease over time (Supplementary Figure 2A). Unilateral sensing animals are not biasing their turn rate in response to the light intensity change over time. However, one-eyed animals show an increased turn frequency (Supplementary Figure 2A) during light increase, decrease and absence of light stimulation. This defect may be due to the lack of binocularity, quantitative loss of PRs or a general side effect of the ablation by itself. No differences were observed between control and one-eyed animals regarding their crawling speed and their head sweep sizes (Supplementary Figure 3). Interestingly, accepted head sweeps are greater in size than rejected ones (Supplementary Figure 3B).

To assess whether larval phototaxis is solely based on temporal information, we performed experiments in an assay using a projector as directional light source (see method section) (Fig. 1b). This setup creates a spatial light gradient, which can be used by larvae to perform navigational strategies based on left–right comparisons, however it may not exclude that a larva is detecting temporal changes in light intensity while moving. We analyzed three different parameters. First, we analyzed the first head sweep direction in a binomial way, which means that we divided first head sweep direction events into two categories: events of larvae swinging their heads toward their left side and events of larvae swinging their heads toward their right side. Second, we tested for the turn direction by dividing all turn events into two categories: turns toward left and turns toward right (in relationship to stimulus and larval heading direction before and after a turn). Third, we determined the mean run change by calculating the differences in larval heading at the beginning and at the end of each run. Steering within runs to the left would be indicated by positive and steering to the right would be indicated by negative values.

When initially heading directly toward a light source, larvae should show no preference to turn left or right, as either direction is away from the light, and control larvae are equally likely to turn left or right[6] (Fig. 2a). For eye ablation experiments we merged the data in an achiral dataset. Larvae with either the left or right eye ablated bias their turn direction to the side of their ablated eye suggesting a spatial comparison of light input between the left and right eye[1] (Fig. 2a). Furthermore, one-eyed larvae bias the direction of their first head sweeps toward the ablated eye (Fig. 2b). Because the direction of the first head sweep is chosen before the head cast, this bias further supports spatial processing of light intensity between the left and right eye[1]. Thus, the first head sweep direction impacts the decision of the turn direction (Fig. 2a, b). To exclude that the ablation has a general effect on these directional biases, we tested one-eyed larvae in complete darkness and analyzed their behavior in the same way. In absence of a light stimulus one-eyed larvae do not bias the direction of their turns nor the first head sweep (Fig. 2a, b).

Another navigational strategy to avoid a light source could be to steer within runs away from it. However, neither bi- nor unilateral sensing animals bias their steering within runs while heading toward the light source (Fig. 2c).

We next analyzed if larval phototaxis benefits from biasing the first head sweep direction when larvae are heading perpendicular to the light source. When heading toward +90° larvae lacking the left eye possess the functional eye on the side where the light source is positioned, while if these animals head toward −90° their functional eye is oriented away from the light source (Fig. 3a). The same is true in an inverted fashion for larvae lacking the right eye (Fig. 3a). Therefore, we have used again eye ablation data for both eyes in an achiral dataset. If larvae use a spatial integration mechanism to avoid a light source, they might bias the first head sweep direction away from the light source and/or steer away from it within runs, when heading perpendicular to the stimulus[6] (Fig. 3b). Thus, we grouped all events when animals had the remaining eye either on the body side toward the light source or away from it (Fig. 3a). Control animals bias their first head sweeps toward the darker side independent whether the light source was to their right (initial heading +90) or to their left (initial heading −90) excluding an effect of handedness in the assay (Fig. 3b). However, larvae with one eye ablated are unable to bias the first head sweep direction when the light is incident on the ablated eye (Fig. 3b).

We next assessed the turn direction of larvae when oriented perpendicular to the light source. We found that bilateral sensing larvae were able to bias the direction of their turns away from the light source. One-eyed animals were biasing the direction of their turns away from the light source, in case the remaining eye was on the body side facing toward the light (Fig. 3c). To control that the observed behaviors were not a general effect of the ablation, we analyzed the behavior of the one-eyed larvae in constant darkness in the same way. One-eyed animals do not show these behavioral biases in absence of a light stimulus (Fig. 3b, c). These findings further highlight the importance of binocularity to perceive spatial information for visual navigation (tropotaxis).

It was recently shown that for chemotaxis larvae also steer during runs by bending the run in a preferred direction[7]. We asked whether steering in runs may also be used as a navigational strategy for larval phototaxis and if it requires a left–right comparison. To avoid the light, larvae, which are heading to +90°, should steer to the left side in runs, whereas in runs toward −90° larvae should steer to the right side. We found that bilateral sensing animals behave in such a fashion, but larvae with ablated eyes are only able to steer away from the light source when the remaining eye is on the light-exposed side of the animal (Fig. 3d). Furthermore, the mean steering within runs in any direction is not modulated by light intensity changes (Supplementary Figure 2B). Thus, both navigational strategies "first head sweep direction" and "steering in runs" are based on spatial integration of light cues. Binocularity could either be necessary for the function of an intensity comparator or for directional eyes. Loss of one eye should lead to a completely disabled intensity comparator. That larvae are able to bias their movements away from the light when their functional eye is on the light-exposed side is therefore suggestive for directional eyes. Binocularity seems to be necessary to sense differences of light intensity spatially. Interestingly, unilateral sensing larvae have a decreased but not completely abolished navigation index (Supplementary Figure 4A). In summary, visually guided navigation of *Drosophila* larvae requires a combination of temporal and spatial information processing.

**Two PR subtypes differentially contribute to navigation.** Previous studies highlighted the impacting role of Rh5-PRs in different navigational paradigms[6,9,13,27]. However, the function of

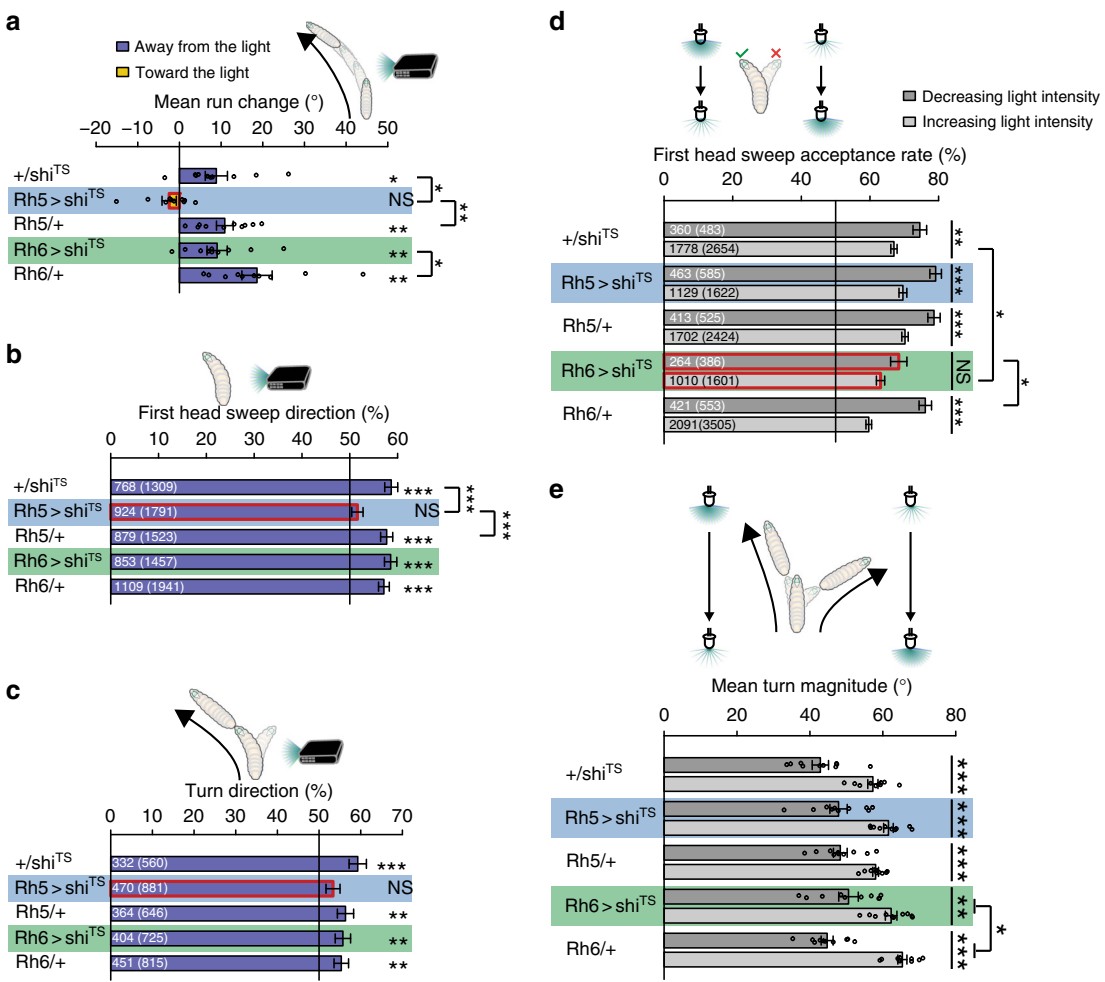

**Fig. 4** Function of the two PR subtypes in phototaxis. **a** Rh5-PRs are necessary to steer away from the light. $+/shi^{TS}$: $n = 10$, $p = 0.011$; $Rh5 > shi^{TS}$: $n = 10$, $p = 0.1844$; $Rh5/+$: $n = 10$, $p = 0.0015$; $Rh6 > shi^{TS}$: $n = 10$, $p = 0.0078$; $Rh6/+$: $n = 10$, $p = 0.0015$. Larvae with silenced Rh5-PRs show a decreased steering away from the light compared to controls. One-way ANOVA: $p = 3.0 \times 10^{-5}$. Dunnett's test: $Rh5 > shi^{TS}$ vs $+/shi^{TS}$: $p = 0.0122$; $Rh5 > shi^{TS}$ vs $Rh5/+$: $p = 0.0024$; $Rh6 > shi^{TS}$ vs $+/shi^{TS}$: $p = 1$; $Rh6 > shi^{TS}$ vs $Rh6/+$: $p = 0.0412$. **b** Functional Rh5-PRs are necessary to bias the first head sweep direction. $+/shi^{TS}$: $n = 1309$, $p = 6.3 \times 10^{-10}$; $Rh5 > shi^{TS}$: $n = 1791$, $p = 0.1857$; $Rh5/+$: $n = 1523$, $p = 2.4 \times 10^{-9}$; $Rh6 > shi^{TS}$: $n = 1457$, $p = 3.7 \times 10^{-10}$; $Rh6/+$: $n = 1941$, $p = 6.3 \times 10^{-10}$. Only the probability of the first head sweep direction of larvae lacking functional Rh5-PRs was statistically different from the controls. $Rh5 > shi^{TS}$ vs $+/shi^{TS}$: $p = 3.7 \times 10^{-4}$; $Rh5 > shi^{TS}$ vs $Rh5/+$: $p = 9.3 \times 10^{-4}$; $Rh6 > shi^{TS}$ vs $+/shi^{TS}$: $p = 0.9692$; $Rh6 > shi^{TS}$ vs $Rh6/+$: $p = 0.5597$. **c** Larvae lacking functional Rh5-PRs are not able to bias the direction of their turns away from the light source. $+/shi^{TS}$: $n = 560$, $p = 6.4 \times 10^{-5}$; $Rh5 > shi^{TS}$: $n = 881$, $p = 0.0506$; $Rh5/+$: $n = 646$, $p = 0.0032$; $Rh6 > shi^{TS}$: $n = 725$, $p = 0.0032$; $Rh6/+$: $n = 815$, $p = 0.0032$. $Rh5 > shi^{TS}$ vs $+/shi^{TS}$: $p = 0.1182$; $Rh5 > shi^{TS}$ vs $Rh5/+$: $p = 0.3369$; $Rh6 > shi^{TS}$ vs $+/shi^{TS}$: $p = 0.3369$; $Rh6 > shi^{TS}$ vs $Rh6/+$: $p = 0.9182$. **d** Functional Rh6-PRs are necessary for accepting more head sweeps during a decrease than increase of light intensity. $+/shi^{TS}$: $n = 3137$, $p = 0.0012$; $Rh5 > shi^{TS}$: $n = 2207$, $p = 1.9 \times 10^{-5}$; $Rh5/+$: $n = 2949$, $p = 1.3 \times 10^{-4}$; $Rh6 > shi^{TS}$: $n = 1987$, $p = 0.0513$; $Rh6/+$: $n = 4058$, $p = 1.6 \times 10^{-13}$. $Rh5 > shi^{TS}$ vs $+/shi^{TS}$ (intensity decrease): $p = 0.1060$; $Rh5 > shi^{TS}$ vs $+/shi^{TS}$ (intensity increase): $p = 0.1264$; $Rh5 > shi^{TS}$ vs $Rh5/+$ (intensity decrease): $p = 0.8829$; $Rh5 > shi^{TS}$ vs $Rh5/+$ (intensity increase): $p = 0.8003$; $Rh6 > shi^{TS}$ vs $+/shi^{TS}$ (intensity decrease): $p = 0.0974$; $Rh6 > shi^{TS}$ vs $+/shi^{TS}$ (intensity increase): $p = 0.0414$; $Rh6 > shi^{TS}$ vs $Rh6/+$ (intensity decrease): $p = 0.0414$; $Rh6 > shi^{TS}$ vs $Rh6/+$ (intensity increase): $p = 0.0548$. **e** Each PR subtype alone is sufficient to perform greater turns during the phase of light intensity increase than during the phase of light intensity decrease. $+/shi^{TS}$: $n = 10$, $p = 8.6 \times 10^{-5}$; $Rh5 > shi^{TS}$: $n = 10$, $p = 1.4 \times 10^{-4}$; $Rh5/+$: $n = 10$, $p = 2.0 \times 10^{-4}$; $Rh6 > shi^{TS}$: $n = 10$, $p = 0.0015$; $Rh6/+$: $n = 10$, $p = 4.3 \times 10^{-8}$. We compared the turn magnitude delta of experimental and control lines. One-way ANOVA: $p = 0.0201$. Dunnett's test: $Rh5 > shi^{TS}$ vs $+/shi^{TS}$: $p = 0.998$; $Rh5 > shi^{TS}$ vs $Rh5/+$: $p = 0.575$; $Rh6 > shi^{TS}$ vs $+/shi^{TS}$: $p = 0.83$; $Rh6 > shi^{TS}$ vs $Rh6/+$: $p = 0.03$. **a–e** In the graphs only statistically significant differences between experimental groups and their respective controls are indicated for simplicity. The data are mean ± SEM. **a**, **e** One sample $t$-test was used. Circles are means of individual experiments. **b**, **c** Exact binomial and **b–d** Fisher's exact test were applied. **a–e** Benjamini Hochberg procedure was used. **b**, **c** The first number gives the number of first head sweeps or turns, respectively, directed to the indicated direction and the number in brackets is the total number of events. **d** The first number is the number of accepted first head sweeps and the number in brackets is the number of all first head sweeps. *$p < 0.05$, **$p < 0.01$, ***$p < 0.001$, $n$ = not significant

Rh6-PRs remains largely elusive. Therefore we next investigated the function of the two PR subtypes for different navigational strategies. To overcome potential developmental defects, we genetically silenced either Rh5- or Rh6-PRs by expressing a temperature sensitive dominant negative form of Dynamin (UAS-$shi^{TS}$), specifically in either PR type. Since an increased

experimental temperature generally changes larval behavior (Supplementary Figure 5), we here compare cross with its proper genetic control at 32 °C[31,32].

We first analyzed if genetically interfering with Rh5-PRs or Rh6-PRs causes deficits in one or more of the five navigational strategies. We found that genetically silencing Rh5-PRs, but not

Rh6-PRs, results in a loss of the ability to steer in runs (Fig. 4a). Similarly, the ability to bias the first head sweep direction and the turn direction away from the light source is lost when silencing Rh5-PRs, but not Rh6-PRs (Fig. 4b, c). Thus, interfering with Rh5-PR function results in the loss of navigational strategies that are based on spatial light cues. Moreover, these findings support that the Rh6-PRs are not essential for spatially encoded navigational strategies (Fig. 4a, b). We next addressed how silencing either PR type affects navigational strategies that depend on temporal light cues, by using the temporal light gradients assay

(see Fig. 1c). If Rh6-PRs are genetically silenced, larvae accept a first head sweep during light intensity increase or decrease with the same probability (Fig. 4d). Control lines (Gal4 or UAS-responder) and larvae with silenced Rh5-PRs are more likely to accept a first head sweep when light intensity is decreasing instead of increasing (Fig. 4d). These results show that Rh6-PRs are in fact required for efficient navigation and that they are necessary for processing of temporal light cues. Since also the turn size is mediated by temporal light information processing we next addressed if this strategy is affected when interfering with PR

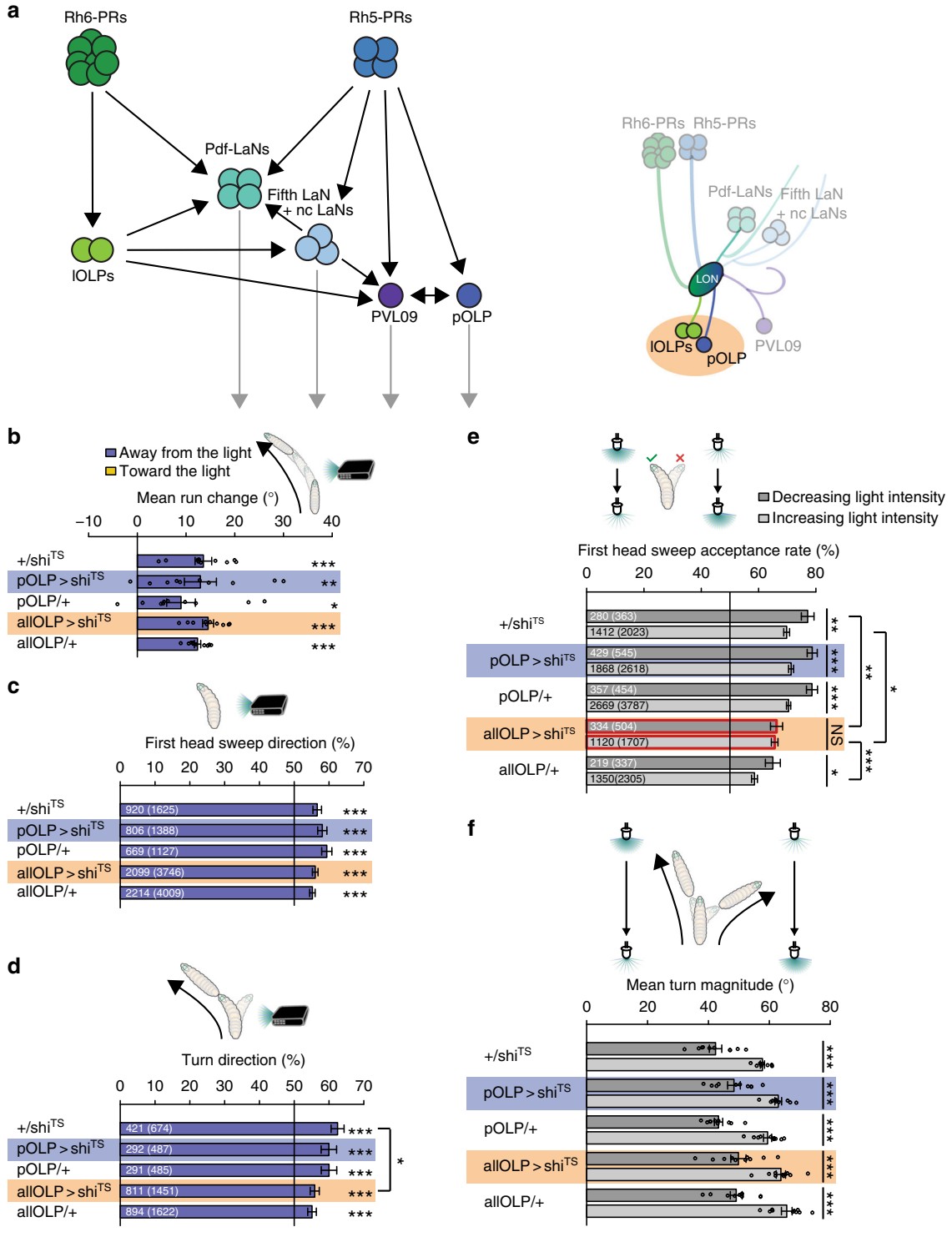

function. Interestingly this navigational strategy was not abolished when either PR subtype is silenced, suggesting that either PR type by itself is sufficient to transmit visual information for this navigational strategy (Fig. 4e). Since silencing Rh6-PRs shows a significant different mean run change and turn magnitude to the Rh6-driver line control, but not to the UAS-*shi^ts* control we cannot exclude a behavioral contribution of Rh6-PRs in these strategies. Interestingly, silencing either PR subtype led to defects in the turn frequency bias and navigation index (Supplementary Figure 6). To further support the role of Rh6-PRs in temporal coding, we performed experiments using yellow LEDs as a directional or temporal light source (Supplementary Figure 7). The published absorption spectrum for Rh6[21], suggests that only Rh6-PRs should detect yellow light, while Rh5-PRs should not be activated. Using yellow light, we do not observe a bias in spatial navigation parameters, while these parameters are biased in blue/green light conditions (Supplementary Figure 7A–D). Moreover, yellow light is sufficient to elicit a significant bias in first head sweep acceptance and turn magnitude, however at a lower rate in comparison to behavior elicit by blue and green light stimulation (Supplementary Figure 7F,G). This experiment further supports the effect of Rh6-PRs on temporal sensing, as we could observe behaviors, which depend on temporal light information processing, by stimulating larvae with yellow light.

Interestingly, phototaxis is not completely abolished when Rh6-PRs silencing is combined with unilateral eye ablation (Supplementary Figure 8). This is most likely due to the fact that these larvae were still able to bias their first head sweep direction and their turn direction to darkness in case the remaining Rh5-PRs were on the side facing the light source (Supplementary Figure 8C–F). The first head sweep acceptance rates and the turn sizes were not biased (Supplementary Figure 8G, H).

In summary, the Rh6-PRs are required for a temporally coded navigational strategy, while the Rh5-PRs are solely essential for spatially coded navigational strategies. Thus, the two PR types take over specific tasks in spatial vs temporal visual information coding.

**OLPs are required for a temporal navigational strategy.** Several distinct types of brain interneurons connecting to the LON have been identified in the past[15,33]. The wiring of the complete visual circuit and the synaptic connectivity between LON innervating neurons has been identified[28]. Two distinct types of PR target neurons can be distinguished: visual local neurons that arborize uniquely in the LON and visual projection neurons that project their axon to defined domains of the central brain neuropil. Interestingly, the local OLPs (lOLPs) (two visual local neurons) predominantly receive input from Rh6-PRs (Fig. 5a). The visual projection neurons include the well-known neurons of the clock circuit, the four *Pigment dispersing factor* expressing lateral neurons (Pdf-LaNs), the fifth lateral neuron (fifth LaN) and two non-clock lateral neurons (nc LaNs)[28]. The projection OLP (pOLP) and the recently identified PVL09 (postero-ventro-lateral neuron 09) also belong to the visual projection neurons[28]. Conversely, visual projection neurons receive direct input almost exclusively from the Rh5-PRs, except for Pdf-LaNs that receive input of both PR types (Fig. 5a).

To first gain insight into the function of visual local neurons, we genetically silenced OLPs. We used two Gal4-driver lines covering either all three OLPs or specifically labeling only the pOLP. When first assessing spatial navigational strategies (first head sweep direction and direction change in run) we found that silencing either all OLPs, the pOLP alone or Gal4/UAS control lines did not result in a significant behavioral change (Fig. 5b, c). These findings suggest that OLPs are not essential for navigation strategies based on spatial information. Larvae show a decreased turn direction bias in comparison to the effector line control in case all OLPs are silenced (Fig. 5d). Further the navigation index was not different from the respective controls (Supplementary Figure 9A).

We next analyzed temporally encoded navigational strategies, when genetically silencing OLPs. We did not observe defects in either navigational strategy when silencing the pOLP alone or in control animals, supporting that the pOLP is not essential for temporal information processing (Fig. 5e, f). However, when silencing all OLPs we observed a defect in the first head sweep acceptance rate compared to the defect observed when Rh6-PRs were silenced (Figs. 4d and 5e). Larvae with silenced OLPs showed similar head sweep acceptance rates for first head swings regardless of whether it was during a period of increasing or decreasing light intensity (Fig. 5e). Since silencing only pOLP does not show a behavioral defect, while silencing all OLPs shows

**Fig. 5** Function of the OLPs in phototaxis. **a** Schematic representation shows the connectome of the larval visual system. Rh5-PRs form chemical synapsis with different projection neurons. Rh6-PRs predominately make synaptic contact with the two lOLPs and the Pdf-LNs. The lOLPs form chemical synapsis with direct target cells of Rh5-PRs (adapted from ref. [28]). **b** OLPs are not necessary to steer away from light. $+/shi^{TS}$: $n = 10$, $p = 3.9 \times 10^{-5}$; $pOLP > shi^{TS}$: $n = 10$, $p = 0.0043$; $pOLP/+$: $n = 10$, $p = 0.0138$; $allOLP > shi^{TS}$: $n = 10$, $p = 1.0 \times 10^{-6}$; $allOLP/+$: $n = 10$, $p = 7.5 \times 10^{-8}$. Mean steering of experimental and control lines is the same. One-way ANOVA: $p = 0.455$. **c** OLPs are dispensable for biasing the first head sweep direction. $+/shi^{TS}$: $n = 1625$, $p = 1.1 \times 10^{-7}$; $pOLP > shi^{TS}$: $n = 1388$, $p = 2.5 \times 10^{-9}$; $pOLP/+$: $n = 1127$, $p = 5.9 \times 10^{-10}$; $allOLP > shi^{TS}$: $n = 3746$, $p = 8.1 \times 10^{-13}$; $allOLP/+$: $n = 4009$, $p = 9.8 \times 10^{-11}$. The probability of the first head sweep direction is not different between experimental and control lines. $pOLP > shi^{TS}$ vs $+/shi^{TS}$: $p = 1.75$; $pOLP > shi^{TS}$ vs $pOLP/+$: $p = 0.687$; $allOLP > shi^{TS}$ vs $+/shi^{TS}$: $p = 0.697$; $allOLP > shi^{TS}$ vs $allOLP /+$: $p = 0.957$. **d** OLPs are dispensable for biasing the turn direction. $+/shi^{TS}$: $n = 674$, $p = 5.1 \times 10^{-10}$; $pOLP > shi^{TS}$: $n = 487$, $p = 1.6 \times 10^{-5}$; $pOLP/+$: $n = 485$, $p = 1.6 \times 10^{-5}$; $allOLP > shi^{TS}$: $n = 1451$, $p = 1.6 \times 10^{-5}$; $allOLP/+$: $n = 1622$, $p = 4.1 \times 10^{-5}$. The probabilities of turning away from the light of larvae lacking functional OLPs and the effector line control is different. $pOLP > shi^{TS}$ vs $+/shi^{TS}$: $p = 0.786$; $pOLP > shi^{TS}$ vs $pOLP/+$: $p = 1$; $allOLP > shi^{TS}$ vs $+/shi^{TS}$: $p = 0.0184$; $allOLP > shi^{TS}$ vs $allOLP /+$: $p = 0.9189$. **e** Functional OLPs are necessary for accepting more first head sweeps during intensity decrease. $+/shi^{TS}$: $n = 2386$, $p = 0.0078$; $pOLP > shi^{TS}$: $n = 3163$, $p = 9.9 \times 10^{-4}$; $pOLP/+$: $n = 4241$, $p = 9.9 \times 10^{-4}$; $allOLP > shi^{TS}$: $n = 2211$, $p = 0.8308$; $allOLP/+$: $n = 2642$, $p = 0.0349$. $pOLP > shi^{TS}$ vs $+/shi^{TS}$ (intensity decrease): $p = 0.8311$; $pOLP > shi^{TS}$ vs $+/shi^{TS}$ (intensity increase): $p = 0.5104$; $pOLP > shi^{TS}$ vs $pOLP /+$ (intensity decrease): $p = 1$; $pOLP > shi^{TS}$ vs $pOLP/+$ (intensity increase): $p = 0.7208$; $allOLP > shi^{TS}$ vs $+/shi^{TS}$ (intensity decrease): $p = 0.0020$; $allOLP > shi^{TS}$ vs $+/shi^{TS}$ (intensity increase): $p = 0.0179$; $allOLP > shi^{TS}$ vs $allOLP/+$ (intensity decrease): $p = 0.8129$; $allOLP > shi^{TS}$ vs $allOLP/+$ (intensity increase): $p = 4.7 \times 10^{-5}$. **f** OLPs are dispensable for making greater turns during intensity increase. $+/shi^{TS}$: $n = 10$, $p = 2.86 \times 10^{-6}$; $pOLP > shi^{TS}$: $n = 10$, $p = 1.03 \times 10^{-5}$; $pOLP/+$: $n = 10$, $p = 6.1 \times 10^{-7}$; $allOLP > shi^{TS}$: $n = 10$, $p = 1.6 \times 10^{-4}$; $allOLP/+$: $n = 10$, $p = 9.27 \times 10^{-6}$. The mean turn magnitude delta differs not between experimental and control groups. One-way ANOVA: $p = 0.818$. **b**, **f** Circles are means of individual experiments. **c**, **d** The first number gives the number of first head sweeps or turns directed to the indicated direction, respectively, and the number in brackets is the total number of events. **e** The first number is the number of accepted first head sweeps and the number in brackets is the number of all events. **b**, **f** One sample *t*-test was used. **c**, **d** Exact binomial and **c–e** Fisher's exact test were applied. **b–f** Benjamini Hochberg procedure was used. The data are shown as mean ± SEM. *$p < 0.05$, **$p < 0.01$, ***$p < 0.001$, n = not significant

a defect suggests that this defect is likely due to the interference with the two lOLPs. Further, silencing all OLPs lead to a decreased bias in turn frequency (Supplementary Figure 9B). Thus, in agreement with the problem in a temporal navigational strategy observed in the Rh6-PRs the two lOLPs appear to function in temporal light perception.

Furthermore, the pOLP alone and all OLPs are not essential for biasing the turn size (Fig. 5f). Interestingly, pOLP is not required for any of the navigational strategies (Fig. 5b–f).

We next investigated the role of other visual projection neurons in phototaxis. We therefore used a Gal4-driver line specifically labeling PVL09, a Gal4 line covering all LaNs (seven cells: four Pdf-LaNs, fifth LaN and two nc LaNs) and a line labeling uniquely the four Pdf-LaNs. Animals in which PVL09 is

genetically silenced are able to perform two spatially encoded phototaxis strategies (first head sweep direction and steering within runs) (Fig. 6a, b). However, the behavioral biases were strongly decreased in comparison with at least one of the respective controls, suggesting that PVL09 could be involved in mediating both behaviors. Further, these animals possess a decreased turn direction bias in comparison with both controls (Fig. 6c). Interestingly, these larvae are also no longer able to bias the acceptance of the first head sweep if the light is decreasing or increasing (Fig. 6d). We were able to further dissect this phenotype. In comparison with the corresponding controls, larvae with silenced PVL09 accept too many first head sweeps while light intensity is increasing (Fig. 6d). The first head sweep acceptance rate is not different from the rate of the controls in

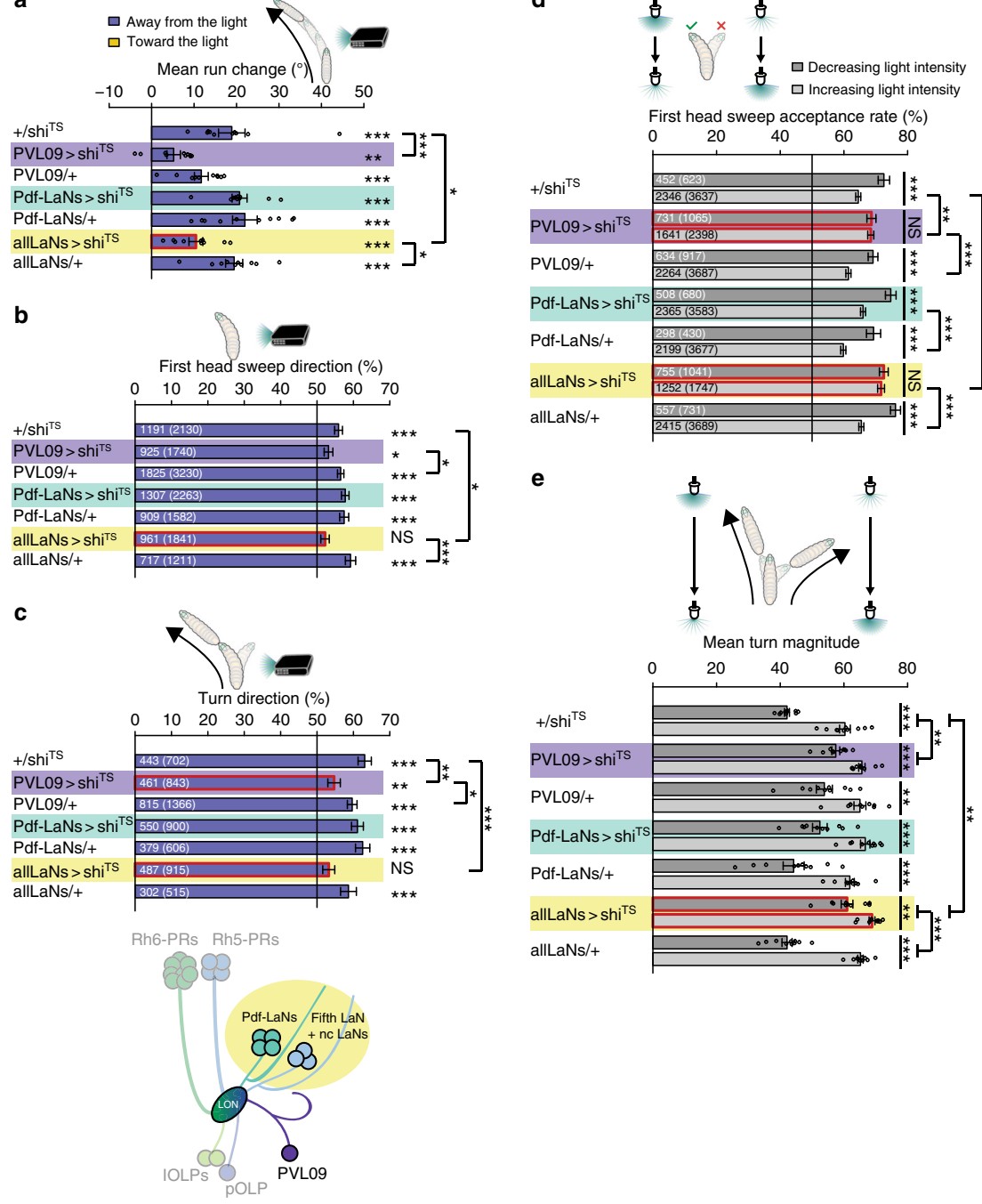

phases of light intensity decrease (Fig. 6d). Thus, PVL09 is critically required in controlling this temporally controlled behavioral strategy. Further, animals with genetically silenced PVL09 show a reduced performance of the other behavioral strategies including the overall navigation index and the turn rate bias (Fig. 6; Supplementary Figure 10).

We next investigated the group of LaNs by either genetically silencing only Pdf-LaNs or all LaNs. Silencing Pdf-LaNs alone did not result in a significant change in any navigational parameter, suggesting that Pdf-LaNs are not essential for visually guided navigation in agreement with previous work[13]. Conversely, genetically silencing all LaNs results in behavioral deficits for all navigation strategies (Fig. 6; Supplementary Figure 10). Interestingly, silencing all LaNs does not completely abolish the spatial navigational strategy steering within runs nor the temporally encoded turn magnitude and rate bias, while the first head sweep direction and turn direction probability as well as acceptance rate remains unbiased (Fig. 6; Supplementary Figure 10B). Larvae in which all LaNs are silenced show a decreased ability to steer within runs and to bias their turn size and frequency as well as a decreased probability to swing the first head sweep and perform the final turn away from the light source compared to these parameters of both control lines (Fig. 6a, b, c, e; Supplementary Figure 10B). Comparing the first head sweep acceptance rates of the phases of light intensity increase or decrease with the corresponding rates of the respective control lines, we observed that larvae lacking all LaNs accept more first head sweeps during light intensity increase than the control lines (Fig. 6d). The first head sweep acceptance rate during light intensity decrease did not differ from the acceptance rate of the control lines (Fig. 6d). The overall navigation index of larvae with all LaNs not functional was decreased in comparison with the respective controls (Supplementary Figure 10A).

Thus, blocking the neurotransmission of Rh5-PRs led to defects in all three navigational strategies based on spatial light intensity processing (Fig. 4a–c). By silencing individual and subgroups of Rh5-target cells, we observed a defect in all navigational strategies (Fig. 6). Interestingly, solely silencing the pOLP or the Pdf-LaNs did not lead to a defect in one of the navigational strategies including the turn frequency bias (Figs. 5 and 6; Supplementary Figures 9, 10).

## Discussion

Animals may use both spatial as well as temporal information about a specific stimulus to most effectively control their navigational decisions. As previously shown, larvae are able to use temporal light cues for negative phototaxis[6]. A first strategy to perceive spatial information is the translation of spatial cues into temporal cues during larval head sweeps. If the animal sweeps toward the light source it would detect an increase in light intensity over time, while when sweeping away from the light source, the larvae would perceive a decrease in light intensity[1,6]. This way spatial light information can be converted in temporal varying signals by self motion, and larvae would orient to the side where the light input becomes the smallest[1,6]. If larvae are also able to perceive spatial visual cues remained unknown.

In most animals, the perception of spatial visual information is achieved by a specialized anatomical organization of PRs in the retina as well as the neural network they contact. The organization of the *Drosophila* larval eye lacks anatomical features that would allow spatial information coding in one eye. However, since larvae possess two eyes located at both sides of the larval head skeleton it was suggested that larvae could use binocular cues by comparing input from the left with input from right eye[1,20].

By studying one-eyed animals, we found indeed that the final turn direction is not only based on this temporal integration, but also to a great extent requires a spatial comparison between both eyes. Thus, while in normal conditions binocularity is not essential for navigation, it is important for certain navigational strategies. The animal uses binocularity to control the direction of the first head sweep away from a light source and for steering in

**Fig. 6** Function of projection neurons in phototaxis. **a** Larvae of all genotypes steer away from light. $+/shi^{TS}$: $n = 10$, $p = 2.2 \times 10^{-4}$; $PVL09 > shi^{TS}$: $n = 10$, $p = 0.0094$; $PVL09/+$: $n = 10$, $p = 9.0 \times 10^{-5}$; $Pdf\text{-}LaNs > shi^{TS}$: $n = 10$, $p = 6.5 \times 10^{-6}$; $Pdf\text{-}LaNs /+$: $n = 10$, $p = 8.0 \times 10^{-5}$; $allLaNs > shi^{TS}$: $n = 10$, $p = 1.8 \times 10^{-4}$; $allLaNs /+$: $n = 10$, $p = 1.7 \times 10^{-5}$. Larvae with LaNs silenced steer less compared to controls. One-way ANOVA: $p = 8.7 \times 10^{-7}$. Dunnett's test: $PVL09 > shi^{TS}$ vs $+/shi^{TS}$: $p = < 0.001$; $PVL09 > shi^{TS}$ vs $PVL09/+$: $p = 0.1825$; $Pdf\text{-}LaNs > shi^{TS}$ vs $+/shi^{TS}$: $p = 0.9821$; $Pdf\text{-}LaNs > shi^{TS}$ vs $Pdf\text{-}LaNs/+$: $p = 0.9973$; $allLaNs > shi^{TS}$ vs $+/shi^{TS}$: $p = 0.0418$; $allLaNs > shi^{TS}$ vs $allLaNs/+$: $p = 0.026$. **b** LaNs are necessary for biasing the first head sweep direction. $+/shi^{TS}$: $n = 2130$, $p = 7.3 \times 10^{-8}$; $PVL09 > shi^{TS}$: $n = 1740$, $p = 0.0104$; $PVL09/+$: $n = 3230$, $p = 5.9 \times 10^{-13}$; $Pdf\text{-}LaNs > shi^{TS}$: $n = 2263$, $p = 5.9 \times 10^{-13}$; $Pdf\text{-}LaNs/+$: $n = 1582$, $p = 5.7 \times 10^{-9}$; $allLaNs > shi^{TS}$: $n = 1841$, $p = 0.0622$; $allLaNs /+$: $n = 1211$, $p = 3.7 \times 10^{-10}$. $PVL09 > shi^{TS}$ vs $+/shi^{TS}$: $p = 0.1372$; $PVL09 > shi^{TS}$ vs $PVL09/+$: $p = 0.0498$; $Pdf\text{-}LaNs > shi^{TS}$ vs $+/shi^{TS}$: $p = 0.2675$; $Pdf\text{-}LaNs > shi^{TS}$ vs $Pdf\text{-}LaNs/+$: $p = 0.8683$; $allLaNs > shi^{TS}$ vs $+/shi^{TS}$: $p = 0.0498$; $allLaNs > shi^{TS}$ vs $allLaNs/+$: $p = 8.9 \times 10^{-4}$. **c** Larvae lacking functional LaNs don't bias their turn direction. $+/shi^{TS}$: $n = 702$, $p = 3.3 \times 10^{-12}$; $PVL09 > shi^{TS}$: $n = 843$, $p = 0.0084$; $PVL09/+$: $n = 1366$, $p = 3.3 \times 10^{-12}$; $Pdf\text{-}LaNs > shi^{TS}$: $n = 900$, $p = 6.4 \times 10^{-11}$; $Pdf\text{-}LaNs/+$: $n = 606$, $p = 1.2 \times 10^{-9}$; $allLaNs > shi^{TS}$: $n = 915$, $p = 0.0551$; $allLaNs /+$: $n = 515$, $p = 1.4 \times 10^{-4}$. Larvae lacking functional PVL09 show a decreased probability to bias the turn direction compared to controls. $PVL09 > shi^{TS}$ vs $+/shi^{TS}$: $p = 0.0027$; $PVL09 > shi^{TS}$ vs $PVL09/+$: $p = 0.0474$; $Pdf\text{-}LaNs > shi^{TS}$ vs $+/shi^{TS}$: $p = 0.5239$; $Pdf\text{-}LaNs > shi^{TS}$ vs $Pdf\text{-}LaNs/+$: $p = 0.5892$; $allLaNs > shi^{TS}$ vs $+/shi^{TS}$: $p = 4.5 \times 10^{-4}$; $allLaNs > shi^{TS}$ vs $allLaNs/+$: $p = 0.0788$. **d** Larvae with silenced PVL09 or LaNs accept first head sweeps during both phases with same probability. $+/shi^{TS}$: $n = 4260$, $p = 1.5 \times 10^{-4}$; $PVL09 > shi^{TS}$: $n = 3463$, $p = 0.9368$; $PVL09/+$: $n = 4604$, $p = 3.0 \times 10^{-5}$; $Pdf\text{-}LaNs > shi^{TS}$: $n = 4263$, $p = 2.7 \times 10^{-5}$; $Pdf\text{-}LaNs/+$: $n = 4107$, $p = 1.9 \times 10^{-4}$; $allLaNs > shi^{TS}$: $n = 2788$, $p = 0.7371$; $allLaNs/+$: $n = 4420$, $p = 6.2 \times 10^{-8}$. Loss of this bias is due to an increased acceptance rate during intensity increase. $PVL09 > shi^{TS}$ vs $+/shi^{TS}$ (intensity decrease): $p = 0.1477$; $PVL09 > shi^{TS}$ vs $+/shi^{TS}$ (intensity increase): $p = 0.0041$; $PVL09 > shi^{TS}$ vs $PVL09/+$ (intensity decrease): $p = 0.9226$; $PVL09 > shi^{TS}$ vs $PVL09/+$ (intensity increase): $p = 2.7 \times 10^{-7}$; $Pdf\text{-}LaNs > shi^{TS}$ vs $+/shi^{TS}$ (intensity decrease): $p = 0.4548$; $Pdf\text{-}LaNs > shi^{TS}$ vs $+/shi^{TS}$ (intensity increase): $p = 0.2428$; $Pdf\text{-}LaNs > shi^{TS}$ vs $Pdf\text{-}LaNs/+$ (intensity decrease): $p = 0.1063$; $Pdf\text{-}LaNs > shi^{TS}$ vs $Pdf\text{-}LaNs/+$ (intensity increase): $p = 2.7 \times 10^{-7}$; $allLaNs > shi^{TS}$ vs $+/shi^{TS}$ (intensity decrease): $p = 1$; $allLaNs > shi^{TS}$ vs $+/shi^{TS}$ (intensity increase): $p = 5.8 \times 10^{-7}$; $allLaNs > shi^{TS}$ vs $allLaNs/+$ (intensity decrease): $p = 0.1477$; $allLaNs > shi^{TS}$ vs $allLaNs/+$ (intensity increase): $p = 1.5 \times 10^{-5}$. **e** Larvae make greater turns when intensity increases. $+/shi^{TS}$: $n = 10$, $p = 3.8 \times 10^{-8}$; $PVL09 > shi^{TS}$: $n = 10$, $p = 1.2 \times 10^{-4}$; $PVL09/+$: $n = 10$, $p = 0.0022$; $Pdf\text{-}LaNs > shi^{TS}$: $n = 10$, $p = 1.4 \times 10^{-4}$; $Pdf\text{-}LaNs/+$: $n = 10$, $p = 1.5 \times 10^{-4}$; $allLaNs > shi^{TS}$: $n = 10$, $p = 0.0015$; $allLaNs/+$: $n = 10$, $p = 1.2 \times 10^{-9}$. Larvae with LaNs silenced show a decreased turn size bias compared to controls. One-way ANOVA of turn magnitude bias: $p = 5.4 \times 10^{-7}$. Dunnett's test: $PVL09 > shi^{TS}$ vs $+/shi^{TS}$: $p = 0.0024$; $PVL09 > shi^{TS}$ vs $PVL09/+$: $p = 0.7852$; $Pdf\text{-}LaNs > shi^{TS}$ vs $+/shi^{TS}$: $p = 0.5408$; $Pdf\text{-}LaNs > shi^{TS}$ vs $Pdf\text{-}LaNs/+$: $p = 0.6846$; $allLaNs > shi^{TS}$ vs $+/shi^{TS}$: $p = 0.0016$; $allLaNs > shi^{TS}$ vs $allLaNs/+$: $p = <0.001$. The data are mean ± SEM. **a, e** One sample *t*-test was used. **b, c** Exact binomial and **b–d** Fisher's exact test were applied. **a–e** Benjamini Hochberg procedure was used. *$p < 0.05$, **$p < 0.01$, ***$p < 0.001$, $n$ = not significant

runs. The quantitative loss of PRs due to ablation is unlikely to explain our results as we could demonstrate that unilateral sensing larvae are able or unable to perform a certain behavior (first head sweep direction bias or steering within runs) with respect to the heading direction (functional eye facing the light source or not). Steering in runs and the first head sweep-direction is controlled by left–right comparison (tropotaxis). Thus, to achieve most efficient phototaxis the animals integrate a combination of different navigational strategies, which are based on temporal and/or spatial information processing. To our knowledge this is a novelty in larval taxes, since previous findings support that thermotaxis and chemotaxis seem to be based uniquely on temporal information[3–5,8,30,34]. Interestingly, adult flies phototaxis is proposed to be based solely on spatial cues[35].

For positive thermotaxis bilateral sensing seems to be unnecessary as larvae do not steer within runs or bias their first head sweep direction[3,8]. Whether these two navigational strategies contribute to chemotaxis remains debated[4,5,7]. Despite these differences, there are shared properties between larval sophisticated navigation strategies among different types of modality taxes. In agreement with our results is the great importance of the first head sweep direction in the different modality taxes[3,4]. Thus, being able to bias the first head sweep direction should be a great advantage for larval navigation. Furthermore, larvae bias the direction and the size of their turns with respect to the stimulus independent of the taxis modality type[3–6,9,36].

These differences and similarities between navigation in response to different modalities raise the question of how independent or shared the sensorimotor programs for each modality in fact are and at what stage neural pathways converge for multisensory integration controlling navigation. Recently published computational models based on behavioral data suggest that at least odor and light information seem to be integrated at early larval nervous system stages[29]. Moreover, visual projection neurons and olfactory projection neurons connect to neurons in the lateral horn region and share some of their target neurons, even though the exact neural networks have not been worked out (Neagu-Maier, Sprecher and Cardona, personal observation). Thus, such circuits would allow convergent cues of distinct modalities controlling avoidance behavior.

Since the larval eye possess two PR types expressing different Rhodopsins with overlapping spectra, the blue-tuned Rh5 or the green-tuned Rh6, this would in principle allow larvae to discriminate colors[21–24]. However, so far Rh5-PRs are required and Rh6-PRs are dispensable for light avoidance behavior under laboratory conditions, even though the wavelengths of the light stimulus ranges from ultraviolet to green, suggesting a different use of two PR types than color discrimination[6,9,13,27].

In line with all the previous studies, we observed an important role of Rh5-PRs in light avoidance. Rh5-PRs are necessary to mediate the two newly described navigational strategies based on spatial information processing (first head sweep direction and steering within runs bias), whereas Rh6-PRs were dispensable for these navigational strategies. Previous studies primarily focused navigation strategies based on temporal light intensity or used simple choice assays. We here test the role of both PR subtypes also using temporal lightscapes as well as directional light gradients. Strikingly, we found that Rh5-PRs are dispensable for navigational strategies that are based on temporal integration. Thus, while Rh5-PRs are important for overall phototaxis they predominantly contribute to navigation based on spatial information. Furthermore, we found that Rh6-PRs in fact function for navigation by controlling if a head sweep should be rather accepted or rejected, a behavioral decision that is dependent on temporal light cues.

That the PR subtypes mediate phototaxis by task sharing of temporal and spatial information may allow better signal-to-noise detection in rapidly varying environmental conditions. We demonstrated recently that on a clear and sunny day larvae are able to navigate away from the sun by the use of either PR type alone, whereas under more diffuse lighting conditions (cloudy sky) both PR subtypes are necessary[9]. Outdoor experiments combined with the use of our tracking system will be of great interest to further address the individual function of the two PR subtypes under distinct lighting conditions. Interestingly, the role of Rh6-PRs in detecting self-induced motion (in the head sweep) displays some similarities to outer PRs in the adult fly retina, where inner PRs are thought to primarily contribute to color discrimination, while outer PRs contribute to motion detection[37–40]. Several similarities of Rh5-PRs with inner PRs and Rh6-PRs with outer PRs can also be found on the anatomical organization of the LON as well as the connectivity of neurons in the LON[23,28,41]. Rh6-PRs predominantly connect to local neurons, that synapse onto Rh5-target neurons, comparable to outer PRs that connect to neurons of the lamina that connect to inner PR targets in the medulla[28,37,39,42]. The fact that *Drosophila* larvae may visually perceive and respond to other moving larvae further suggests a temporal detection mechanism of rapid light changes produced by other animals[16,18,19]. Moreover, larvae are able to use temporal visual cues to avoid collisions with each other[19]. It was also reported that larvae are visually attracted by wiggling larvae when these are glued to cover of the petri-dish arena[16,18]. It seems possible that for these types of behavior Rh6-dependent temporal cues may be required.

However, the task sharing between PR subtypes to transmit temporal and spatial cues does not explain the need of two different Rhodopsins. In theory, a single PR subtype would be sufficient for the navigational decisions that we described. A possible advantage of having a blue- and green-tuned PR type is based on the distribution of colors of celestial light. On a sunny day, the relative intensity of light is max for blue[43,44]. However, under cloudy conditions the color content of skylight is shifted toward longer wavelengths (e.g. green)[43,44]. The same is true for light transmitted through leafs[44]. Under ambiguous lighting situations, it could be beneficial to perceive and process light of longer wavelengths (green) over shorter wavelengths (ultraviolet to blue). Thus, it is more likely that the use of Rh6 instead of a different Rhodopsin is not essential for the function of this PR subtype in general, but that it may rather be relevant as adaption to the lighting conditions under which the task of these PR subtype is most essential.

Within the visual circuit the Rh6-PRs pathway is converging with the projecting Rh5-PRs pathway already at the level of the second synapse[28]. Rh6-PRs predominantly connect to the two lOLPs, which in turn locally connect to visual projection neurons[28]. Moreover, both lOLPs are reciprocally interconnected[28]. Considering these wiring properties of lOLPs toward visual projection neurons and the behavioral observation when manipulating the OLPs, we hypothesize that lOLPs are involved in temporal processing of Rh6-input, which they further transmit toward visual projection neurons[28]. Genetically blocking the neurotransmission of all OLPs results in the same behavioral phenotype as the blocking of Rh6-PRs: the loss of differentially regulating head sweep acceptance if light increases or decreases. Thus, the temporal defect when blocking the neurotransmission of Rh6-PRs may be due that lOLPs lack visual input. Since the two lOLPs express either a putatively excitatory neurotransmitter (acetylcholine) or inhibitory transmitter (glutamate), it seems likely that they transmit both excitatory and inhibitory information during light increment or decrement[28]. In such a fashion changes in light intensity could be enhanced independent of

direction (increase or decrease of light intensity)[28]. Furthermore, such a module could be able to provide scale-free detection of light intensity changes if it would purely respond to changes in light intensity and not to constant light itself[28]. This would be a great advantage for the decision-making process to accept or reject a head sweep under ambiguous lightscapes, with a bad signal-to-noise-ratio and/or only faint changes in light intensity. Future experiments on the physiology of the lOLPs will be of great interest to further dissect the precise activity of these intriguing local neurons.

Since most visual projection neurons could receive direct input from Rh5-PRs as well as indirect input from Rh6-PRs (via lOLPs), we would expect to observe defects in both temporally and spatially controlled navigational strategies when silencing visual projection neurons. Interestingly for the Pdf-LaNs, the only set of visual projection neurons that receives direct input from both PR types, we did not observe any behavioral defect in agreement with previous studies that show that Pdf-LaNs are required to modulate light avoidance in a circadian fashion[12,13]. Moreover, when manipulating all LaNs we observed behavioral defects in all navigational strategies, suggesting that the two nc LaNs and the fifth LaN could be involved in controlling all behavioral strategies. Overall it appears that also silencing PVL09 had similar effects for all the navigational parameters. It seems plausible that all LaNs and PVL09 possess similar functional roles and contribute to the performance of all measured navigational strategies. Future physiological experiments on these circuit motifs will probably help to address further the function of the different interneuron subtypes. Thus, transmission of temporal and spatial information to control distinct navigational strategies can be followed to some degree into the LON circuit. Expectedly, here neurons contribute to distinct aspects of navigation presumably by further processing spatial and temporal cues. It will be of great interest to further identify how the Rh6-PR–OLP pathway functions temporal coding and how visual projection neurons allow the transmission of spatial and temporal computations.

## Methods

**Fly strains**. We obtained some of the transgenic fly strains from Bloomington Drosophila Stock Center and HHMI Janelia Farm Research Campus split-Gal4 (SS) collections as indicated and used standard genetics of *Drosophila* to combine the different transgene. We used wildtype *Canton-S*, w[1118] (courtesy of R. Stocker), w[1118]; *Rh5*-Gal4 / CyO,Tb[1]; (Bloomington stock # 7458 # 36335), w[1118];;*Rh6*-Gal4 / Tm6b (Bloomington stock # 7464), w[1118]; *allOLPs*-Gal4 / Tm6b (Bloomington stock # 48306)[28], w[1118]; *pOLP*-Split-Gal4 (SS01745)[28,45], w[1118]; *PVL09*-Split-Gal4 (SS00671)[28,45], UAS-*shi*[TS] attp5[46], UAS-*shi*[TS]:GFP[46], yw; *GMR*-GFP[47], w[1118]; *Pdf*-Gal4 (Bloomington stock # 6900), w[1118]; allLaNs-Gal4 (Bloomington stock # 48842)[28]. All Gal4-driver lines were crossed to UAS-*shi*[TS] effector line. For heterozygous control experiments Gal4-driver and UAS-effector lines were crossed to w[1118]. The neuroanatomy of the lines *allOLPs*-Gal4, *pOLP*-Split-Gal4, *PVL09*-Split-Gal4, and allLaNs-Gal4 was recently described in high detail[28]. Larvae were raised at 25 °C in a 12 h:12 h light–dark cycle on cornmeal medium supplemented with molasses, fructose and yeast. Early larval stage 3 *D. melanogaster* larvae were used for all behavior experiments.

**Preparation of behavioral experiments**. Behavioral experiments were prepared and performed under red light conditions during the day. At least 20 min prior to the start of behavioral experiments larvae were kept in darkness. Additionally, for experiments in which neurotransmission was blocked by inducing high temperature larvae were kept for at least 10 min in a water bath at 32 °C. Larvae were taken out of a food vial with a spoon and cleaned with tap water. Larvae were kept in a water drop at room temperature or at 32 °C for 5–10 min prior to the experiment. Thirty larvae were placed circular in the middle of a 24.5 × 24.5 cm petri dish (BD Falcon BioDishXL, BD Biosciences) filled with 2% agarose (Agarose Standard, Roth). To increase the contrast a black aluminum plate was placed underneath the agarose. The agarose plate was either at room temperature or at 32 °C.

**Tracking system**. The testing plate was placed into a dark box and four rows of red LEDs (623 nm, Conrad) were placed around the agarose plate for equally

illumination. Larvae were allowed to move freely on the plate for 11 min. During this time, a camera (acA2500-14gm, Basler AG, Germany) with a Fujinon lens (Fujinon HF12.5HA-1B 12.5 mm/1.4, Fujifilm, Switzerland) and a light red bandpass filter (BP635, Midwest Optical Systems, USA) was recording the larval behavior with 13 frames/s. A 1 cm checkerboard picture was used for calibration. A customized software developed with LabVIEW was used for image acquisition[5,6]. The customized MAGAT Analyzer software was used for analyzing larval behavior[5,6]. Further analysis, statistical tests and figures were made with MATLAB and R. The first minute of each experiment was not taken into account as larvae were allowed to acclimatize to the behavioral setup.

**Visual stimulation**. We presented different types of visual stimuli to the larvae. In the directional light source assay a projector (EX7200 Multimedia Projector, EPSON) with a colored glass bandpass filter (335–610 nm, BG40, Thorlabs) was placed on one side of the agarose plate. The projector was positioned with an angle of 40°, 26 cm in height and 38 cm in distance away from the center of the agarose plate. At the beginning of the agarose plate the light intensity was 4331 µW/cm$^2$ with two maximum intensity peaks: The first at 438 nm (71.6 µW/cm$^2$) with a half width of 9 nm and the second at 549 nm (47.9 µW/cm$^2$) with a half width of 10 nm.

For the temporal light gradient assay a blue and a green LED (PT-120, Luminus, Billerica, MA, USA) were placed vertical 45 cm above the plate. The highest intensity was 378 µW/cm$^2$ with a first intensity peak at 455 nm (11.9 µW/cm$^2$) with a half width of 9 nm and with a second intensity peak at 522 nm (3.7 µW/cm$^2$) with a half width of 14 nm. Controlled by an Arduino running a customized script, the light intensity of the LEDs was changing over time. For 25.5 s the light intensity was increasing linearly followed by a 4.5 s phase of constant highest light intensity. After this for 25.5 s the light intensity was decreasing linearly followed by a 4.5 s phase of constant lowest light intensity. This cycle was repeated 11 times per experiment. During the phases of intensity change, every 100 ms the intensity changed by approximately 1.5 µW/cm$^2$. The phases of constant light intensity ±1 s were not taken into account as solely larval behavior should be analyzed during phases, in which the animals sense linear increasing or decreasing intensity, respectively. We also performed additional experiments, in which we doubled the speed of intensity change (every 50 ms the intensity changed by 1.5 µW/cm$^2$) or reduced the speed of intensity change by half (every 200 ms the intensity changed by 1.5 µW/cm$^2$) (Supplementary Figure 1). In these temporal assays larva could detect within 1 s intensity changes of 30, 15 and 7.5 µW/cm$^2$, respectively. For the directional light source assay, we measured an average intensity change per cm of 103 µW/cm$^2$. Assuming a larva, which runs straight toward the light source with a speed of 4 cm/min (Supplementary Figure 5E), could detect within 1 s a light intensity change of 6.87 µW/cm$^2$. Thus, the slowest temporal intensity ramp would match best to what a larva could sense when it is running straight toward the light. However, in case larvae have directional eyes (our data suggest this) the light intensity changes might be greater than this.

Further, we used also yellow LEDs (FinePitch LED Strip yellow, Solarox) for both assays. The highest intensity was 183 µW/cm$^2$ with an intensity peak at 593 nm (9 µW/cm$^2$) with a half width of 7 nm. For control experiments, we used the blue and green LEDs mentioned above with highest intensity of 183 µW/cm$^2$. In the temporal assay of these experiments, the light intensity was changing by 0.72 µW/cm$^2$ every 100 ms.

**Laser ablation**. For laser ablation, we used VisiScope CSU-W1 inverted Spinning Disk Confocal Microscope Nikon Ti-E equipped with a laser ablation module (MICROSHIP laser 355 nm passively Q-switched for average power of 16 mW, delivering 2-kW peak power at repetition rates of 21 kHz) connected to a VisiFRAP-DC scanner. We gave laser pulses not longer than 5–10 ms. We used newly hatched yw[1118]; GMR-GFP larvae to locate the Bolwig organ more easily. During laser ablation larvae were physically immobilized on a cover slip by a block of 10% agarose. After Bolwig organ ablation larvae were allowed to recover for three to four days. At least 2 h before behavioral experiments, we verify with a fluorescence stereomicroscope that the Bolwig organ and nerve were not recovered. Larvae were separated in two groups by the criterion left or right Bolwig organ ablated. In contrast to the other behavioral experiments less than 30 larvae were tracked simultaneously. Further, the same one-eyed animals were tested in the directional assay, in the temporal assay and in constant darkness. Beside this, behavioral experiments were performed like described above.

**Navigational parameters**. Larval position, contour and midline and bearing of body, position of head and tail were identified from the recordings produced by the tracking system by the use of a custom machine vision software[5,6,9,48]. Raw data are created as series of images with 13 frames/s. Based on user-specific threshold settings the software removes the dark background from every image and keeps bright spots. These bright spots are fused to single larva. Next, this larva is either forming a new or added to an existing track. Tracks terminate if they leave the field of view of the camera or if two tracks converge indistinguishable into each other[5]. For further analysis, the data are loaded into MATLAB. The custom software called MAGAT Analyzer is partition each track into runs and turns[5]. Runs were defined as forward locomotion, in which larval head and body were aligned. Turns were

defined as periods of slow or no forward locomotion paired with head sweeping behavior. Pauses were cases of slow or no forward locomotion, in which the larval head and body were aligned. The body bend angle was determined by the angle of the two lines, which fit best the animals body midline. The threshold for head and body alignment was set to 20° in consistency with previous studies[5,6,9,48]. In other words, a head sweep initiation is flagged when the body bend angle is greater than 20°. A head sweep ends when this angle is less than 10° or the head swings to the other side of the body or a new run starts. Therefore, also turns or pauses end when a run starts. A head sweep which is followed by a run is called "accepted", whereas a head sweep which is followed by another head sweep or followed by a head–body alignment without forward locomotion is called "rejected". The threshold speed for separating runs from turns/pauses was calculated individually for each larva. These calculations use data immediately before and after turning/pausing. Is larval speed slower than the mean larval velocity immediately before and after turns/pauses, than these events are identified as turns or pauses with respect to the threshold for head–body alignment mentioned above[5,6,9]. The threshold for minimum run time was set to 1 s. All the runs and turns/pauses of an animal were merged to a track. The minimum track length was set to 0.5 cm and the minimum track duration was set to 30 s. A more detailed description can be found elsewhere[5].

For the analysis of the behavioral data produced with the directional light source assay, we used a navigational compass. According to this compass, heading toward the light source is heading to 0° and heading away from the light source is heading to 180°. To the left values are becoming more positive, whereas to the right values are becoming more negative. Heading to +90° larvae are perpendicular to the light source and their right body half is facing toward the light source. Also, larvae are heading to −90° are orientated perpendicular to the light source, but in this case their right body half is pointing away from the light source.

We divided the directions of the compass into four bins each of 90° in size. Like this all the directions from <+45° until >−45° were in the bin called toward the light source (0°, +x). In the bin called away from the light source were all the directions which were >+135° or <−135° (180°, −x). Perpendicular to the light source were all directions which were >+45° and <+135° (+90°, +y) as well as all directions which were <−45° and >−135° (−90°, −y).

We calculated a navigation index, which is the mean velocity in x-direction divided by the mean run speed in all directions. In case all animals would uniformly travel to the light source, this index would be +1. Instead it would be −1, if all larvae would move away from the light. If the animals would navigate unbiased the navigation index would be 0.

Larvae bias their turn direction away from the light source, when they were previously heading perpendicular to the light source[6]. Only turns that followed a previous heading direction perpendicular to the light source were taken into consideration (+90° bin and −90° bin). Previously heading to the +90° bin a clockwise turn would be toward the light and a counterclockwise turn would be away from the light source. In contrast to this when previously heading to the −90° bin a counterclockwise turn would be toward and a clockwise turn would be away from the light source.

To analyze the turn direction of larvae heading toward the light source, we took into account only turns with previous heading to the 0° bin.

With this current work, we described a new navigational parameter for larval phototaxis. Runs with mean heading to +90° or −90° were counted as runs perpendicular to the light source. For runs with mean heading to +90° and −90° we calculated respectively for both bins the mean run change in degree. For each of these runs, we calculated the difference in heading direction between the start and the end of each run. More heading toward the left would be indicated by a positive value and a heading change to the right side would be indicated with a negative value. So, the mean heading change in run was calculated from this values for all runs of the +90° and the −90° bin, respectively, or together (as indicated in the corresponding figures). To analyze the directional change within runs of larvae heading toward the light source, we took into consideration solely runs with previous heading to the 0° bin.

For the navigational parameter, called first head sweep direction, we used two different definitions, because an increase of environmental temperature (needed for experiments using shi[TS] as an effector) led to a decreased number of first head sweeps. For experiments performed at room temperature, we analyzed only first head sweeps that followed a previous heading direction perpendicular to the light source (+90° bin and −90° bin). For experiments performed at 32 °C, we divided the directions of the navigational compass in two bins with each 180° in size. All the positive directions were considered as belonging to the positive half disk of the navigational compass (heading between >0° and <+180°). Whereas all the negative directions were counted to the negative half disk of the navigational compass (heading between < 0° and > −180°). We took only the first head sweeps in consideration. Previously heading to the positive half disk of the compass a head sweep to the left would be away and a head swing to the right would be toward the light source. If previously heading is orientate to the compass half disk with the negative values a head swing to the left would be toward and a head swing to the right would be going to be away from the light source. Both definitions led to comparable results regarding the first head sweep direction of unilateral sensing larvae (Fig. 3b, Supplementary Figure 4B).

For analysis of behavioral data generated with the temporal light source assay, we were taken into consideration first head sweeps and turns, which were performed during the phase of light increase and the phase of light decrease, respectively. Head sweeps are defined as rejected, in case they are followed by another head swing, or defined as accepted, if they are followed by a new run. We calculated the probability that first head sweeps got accepted not rejected for both phases respectively.

The turn size was defined as the difference between the previous heading direction (before the turn) and the new run start direction (after the turn). We calculated a mean turn size for each phase for each experiment separately. Finally, we calculated an overall mean out of the individual means. In order to compare the turn size bias between different groups, we were comparing the turn size deltas. We calculated a turn size delta by subtracting for each experiment the mean turn size of turns occurring in the phase of light intensity decrease from the mean turn size of events in the phase of intensity increase.

For calculating the mean turn rate, the number of all the turns performed during a phase of light intensity change were counted. This number was processed to turns/min and divided by the average number of animals on the plate during the respective phase of intensity change. Finally, a mean was calculated for each experiment. An overall mean was calculated from the individual means of each experiment. In order to compare the turn frequency bias between the different groups, we were comparing the turn rate deltas with each other. The turn rate deltas were calculated similar to the turn size deltas by subtracting for each experiment the mean turn rate of events from the phase of decreasing intensity from the mean turn rate of increasing intensity.

The mean steering in any direction within runs was defined as the mean steering within runs (independent of direction), which occurred during the phase of light intensity increase or decrease. The differences angle between larval heading before and after each run was used for this calculation, but we did not distinguish between steering to left or to the right. A mean was calculated for each experiment and an overall mean was calculated from these individual means.

The mean run speed was calculated by dividing each length of a run by its duration. Finally, the mean of all run speeds was determined per experiment. Out of the individual means an overall mean was calculated.

For the mean head sweep size the head sweeps were divided in accepted and rejected head sweeps. The size of a head sweep was determined by the difference in degree between the previous heading direction and the maximal heading direction of a certain head sweep. Means for each experiment were calculated and from these means an overall mean was determined.

**Statistical analysis**. For experiments 30 animals per experiment were used with a repetition of ten experiments per condition. The data are shown as means and the error bars indicate ±SEM. Circles indicate the means of the individual experiments. In case of binary choice data (first head sweep direction, turn direction and first head sweep acceptance rate) we present two numbers. The first number is the number of first head sweeps or turns, which are directed away from the light and the second number in the brackets gives the total number of events. For the head sweep acceptance rate the two numbers are respectively the number of accepted first head sweeps and the total number of first head sweeps, which occurred in the respective phase. SEM of binary choice data was calculated as: (1)

$$\frac{\sqrt{probability \times (100 - probability)}}{\sqrt{total\ number\ of\ events}}$$

Standard statistic functions in MATLAB (MathWorks) ("t-test" and "t-test 2") and in RStudio (RStudio, Inc.) ("binom.test", "fisher.test", "aov" and"glht (multcomp)") were used for performing statistical analysis. A two-tailed one sample t-test was performed to test the navigation index and the mean run change of the different datasets respectively against chance. A two-tailed two sample t-test was used to test the respective means of intensity increasing and decreasing phase against each other for the turn magnitude bias, the turn rate bias and the steering within runs in temporal assays. A two-tailed two sample t-test was used to test the mean head sweep sizes of accepted and rejected head sweeps against each other. A two-tailed two sample t-test was performed to test the mean speed of different groups against each other. A one-way ANOVA followed by Dunnett's test was performed to test the navigation index, the mean run change, the mean turn magnitude delta and the mean turn rate delta for each experiment of the different datasets, respectively, against the particular values of the corresponding effector and driver line controls. A two-tailed exact binomial test was performed to test the probability of first head sweep direction and the probability of turn direction of the different datasets, respectively, against chance. A two-tailed fisher's exact test was performed to test the distribution of first head sweep direction and the distribution of turn direction of the different datasets, respectively, against the particular distributions of the corresponding controls. A two-tailed fisher's exact test was also performed to test of a particular dataset the distribution of first head sweep acceptance and rejection of time phases of light intensity increase against the phases of light intensity decrease. Furthermore, a two-tailed fisher's exact test was used to probe, if the distribution of first head sweep acceptance and rejection of a particular dataset and a corresponding control is the same with respect to the phase of light intensity increase and decrease. Rejection of the null hypothesis that the means or distributions of events of datasets are the same or that a mean or distribution of events of a dataset is chance: *p < 0.05, **p < 0.01, ***p < 0.001. We performed the Benjamini Hochberg procedure in cases of multiple testing, to adjust

*p*-values. Exact OR-values and degrees of freedom as well as for *t*-tests *t*-values and for ANOVAs *F*-values are provided in the Supplementary Tables 1 and 2.

**Code availability**. Please find the MAGAT Analyzer software package here: https://github.com/samuellab/MAGATAnalyzer.

**Data availability**. All relevant data are available from the authors upon reasonable request.

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

## Acknowledgements

We would like to thank the Bioimage Light Microscopy Facility of the Departments Medicine and Biology of the University of Fribourg for their great support. We would like to thank Gerald M. Rubin for providing unpublished reagents and the Bloomington Drosophila Stock Center for fly stocks. Furthermore, we thank Rudolf Rohr for giving us advices and suggestions for the statistical analysis. We also thank F. Javier Bernardo-Garcia, Pauline M. Fritsch and our other colleagues at the University of Fribourg for fruitful discussions and comments. This work was funded by the Swiss National Science Foundation (31003A_149499 and 31003A_169993 to S.G.S.), the European Research Council (ERC-2012-StG 309832-PhotoNaviNet to S.G.S.), the National Science Foundation (NSF 1455015 to M.G.) and the National Institutes of Health (NIH 1DP2EB022359 to M.G.).

## Author contributions

T.-H.H. and S.G.S. designed the experiments and wrote the manuscript. T.-H.H. performed the experiments and analyzed the data. P.B., B.A., M.Z., J.W.T., M.G. and A. S. provided new reagents or technical support. S.G.S. designed and coordinated the study.

## Additional information

**Competing interests:** The authors declare no competing interests.

