## [Peer Review File · Nature Communications]

Reviewers' comments:

Reviewer #1 (Remarks to the Author):

This paper examines negative phototaxis in *Drosophila* larvae. Using two different light presentation paradigms, the authors demonstrate that larvae can use both spatial and temporal cues to bias navigation behavior. Consistent with the idea that larvae use spatial cues for navigation, ablation of either the left or right eye interfered with putative spatial navigational strategies. Finally, the authors used genetic silencing experiments to dissect the role of photoreceptors and LON neurons in these two strategies, and found distinct roles for different neuron types.

Overall I found this paper very interesting. The topic of navigation is of great interest to many researchers, and the dissection of visual navigation into spatial and temporal components is novel. My enthusiasm was lessened somewhat by a difficult exposition. In particular, the methods used to analyze the data are not adequately documented, and insufficient detail is provided in the main text and legends in order for a reader to easily evaluate the material presented in the figures. The paper could also be strengthened by the addition of some controls, and by further experiments looking at the interaction of spatial and temporal navigation modes.

Major concerns:

- 1) The Methods do not adequately document the criteria for identifying head sweeps, runs, etc. No quantitative criteria are given for these designations, rather a previous paper is referenced. Quantitative criteria must be included, as well as some sense of the degree to which the data could be easily parsed into these different categories.
- 2) A shorthand description of how each of the measures shown in the figures was calculated (eg “delta of mean turn magnitude 0° vs 180°”) should be provided in the main text and/or figure legend, at least for the first case of each of these appearing. Without this it is very difficult for a reader to evaluate the figures.
- 3) The authors should provide some control for the effects of ablation on navigation and movement. A sham lesion of another group of neurons would greatly strengthen the results in figure 2. The authors should also analyze whether unilaterally lesioned larvae show any differences in the frequency or magnitude of head sweeps.
- 4) The effect of silencing Rh6 neurons on temporal sensing is not very striking (figure 4C). Can the differences between Rh5 and Rh6-mediated behavior be strengthened by performing temporal or spatial experiments with narrowband light stimuli?
- 5) It would be interesting to look at the relationship between temporal and spatial navigation strategies. For example, how is phototaxis impaired when Rh6 or iOLP silencing is combined with unilateral eye ablation? Does this completely abolish phototaxis?
- 6) The distinction between the effects of PVL09 silencing and allLaN silencing (figure 6) seems a bit forced. Overall it appears that these two manipulations had similar effects for all parameters measured, however, the PVL09 controls also had reduced behavior, making the difference not statistically significant. It seems plausible that both of these groups of neurons play similar roles—something that will probably be resolved by physiology.
- 7) Previous studies have looked at the effect of temporal changes in light intensity on turn rate and not just percent of first head sweeps accepted. How is turn rate affected by the various manipulations performed in this study?

Minor comments:

- 1) It would be helpful to include a reference to the diagram in figure 5 when describing the anatomy of of the LON in the introduction.
- 2) intro line 94: "both navigational strategies are proposed to be mediated by temporal processing" It would be helpful to explain what is meant by temporal processing.
- 3) Previous studies have mostly concluded that larvae use temporal processing for navigation and have also found that Rh5 are the main photoreceptors involved. Yet this study shows that Rh5s are mostly involved in spatial processing. This should be addressed in the discussion.
- 4) Discussion line 427-430. There is a great deal of evidence for temporal processing in olfactory navigation in adult flies.

Reviewer #3 (Remarks to the Author):

1. The authors examine how binocular vision and two photoreceptors mediate the navigational functions of the *Drosophila* larva.
2. The problem that the authors are tackling is very interesting.
3. Figure 1D and Results lines 161-200. The authors present a key result, that the direction of temporal light changes induces a bias in larval turning, whereby diminishing light increases the acuteness of turning. The data show clearly that larvae are able to detect temporal changes in brightness.
4. Figure 1D and Methods lines 624-637. For the light gradient over time, what is the justification to use a 23.5 s gradient?
5. Figure 1D. How does this rate of brightness change compare to the brightness change that a larva would experience during a head sweep?
6. Figure 1D. Have the authors estimated (by calculation) the expected brightness change in a head sweep in the gradient setup?
7. Figure 1D. How does this rate of brightness change compare to the brightness change that a larva would experience during a run?
8. Figure 1E. The axis appears to be mislabelled "0° vs 180°." In the legend, there is mention (lines 807–813) of a yellow and purple compass, but there is no corresponding image in Figure 1.
9. Figure 1E. A delta is reported. However, the raw observed angle ° values are not. Why were the raw ° values not reported?
10. Figure 1E. Why were the angle ° values in the temporal gradient not directly compared with the angle values in constant light or no light from the experiment in Figure 1C? This would allow the reader to contextualize the angles of turning.
11. Figure 1E. The cartoon is somewhat confusing, could be made clearer by having the larva side-by-side rather than back-to-back, and by putting the actual turn angle values next to the larva drawings, perhaps with an angle vertex.
12. Figure 1F. At first glance, there appears to be a large difference in head-sweep acceptance rate. From the relative lengths of the bars, it seems that larvae are greatly more likely to accept a head sweep if it occurs during decreasing light. However, the axis has been truncated below 50%, so the difference is ~62% versus ~51%, a difference of only 11%. I recommend that either (1) truncated axes are removed from the manuscript or (2) the bars in such cases are replaced by point markers (e.g. circles) instead to indicate the mean.
13. Figure 1F and all subsequent Figures. The use of bars-only goes against Nature Press' recommended practice to show all individual observed values ["Show the dots in plots." Nature Biomedical Engineering, 1(5), 2017].

14. Figure 1F. The axis orientation is reversed relative to Figure 1E. I can see no reason to do this, and makes it confusing for the reader. Convention would suggest placing zero on the left.
15. Figure 1C and 1F. The use of a pie chart in one and a bar chart in the latter inhibits the reader from making a comparison between head-sweep acceptance rates for the four conditions.
16. For Figure 1F and all subsequent charts showing % values for a binary choice, the authors used the two-tailed exact binomial test. For these data, how was the error bar SEM calculated?
17. Figures 2-3. The authors present a second key result: that larvae use both eyes to navigate with respect to light. Having both eyes intact is relied upon by several navigational motor functions: head sweep bias, turning direction, and run changes.
18. Figure 2 shows convincingly that larvae favor turns in the direction of an ablated BO, by about 15%, and that this largely derives from a first head-sweep direction in favor (by about 12%) of the ablated eye.
19. Figure 2B. The axis is unlabelled. From the legend, it should be "Navigation Index."
20. Figure 2B. The Navigation Index has no negative control. For example, what is the Navigation index for larvae when the directional source is turned off, i.e. under red light-only conditions?
21. Figure 2B. More generally, no locomotion data are shown. For the BO-ablated animals, is the locomotion (turning metrics, speed) normal?
22. Figure 2C. Truncated axis, but since it is centered around 50%, I recommend replacing bars with mean point markers and dots.
23. Figure 2D. Axis truncation which should be fixed, preferably by extending to zero.
24. Figure 2E. As per 2C.
25. Figure 2. The diagram of the ablated BO is drawn as though it is on the left eye. However these data are labelled "towards the ablated eye," suggesting that data from both left- and right-ablated animals was used, and combined into an achiral data set. If this is correct, then the authors need to mention this somewhere. It is of importance since the control data are chiral, as are the data in Figure 3C.
26. In Figure 3, the authors show that: (1) the first head-sweep direction (FHSD) bias is impaired when an ablated BO faces towards the light; (2) that BO ablation reduces turn direction bias; (3) that run changes become more obtuse when the ablated BO is facing the light source; and (4) light-avoiding maneuvers are close to normal when the intact BO faces the light.
27. On lines 263–265, the authors state "That larvae are able to bias their movements away from the light when their functional eye is on the light-exposed side is suggestive for directional eyes." Two models for binocular function in this case could be (1) an intensity comparator or (2) directional eyes. The intensity comparator should be incapacitated with removal of one eye, since no comparison can be made. A single directional eye should have 100% function in its relevant direction. As the larvae have close to normal FHSD and run change acuity functions on the intact BO side, the authors' data would support the latter model for $\pm 90^\circ$ navigation.
28. Figure 3C. Larvae 1st head-sweep direction (FHSD) bias is unaffected when the ablated eye is distal to the light source, but is eliminated when the ablated eye is proximal to the light source. As 50% is the value of indifference, I recommend replacing bars with point markers.
29. Figure 3C. Do the authors have reason to believe that Drosophila larvae have a handedness in their light avoidance? Otherwise, why is it necessary to analyze the head-sweep direction independently for left-ablated and right-ablated BO? Would it make sense to make an achiral analysis (as appears to have been done for Figure 2) with respect to the light-proximal and light-distal ablated BO instead?
30. Figure 3D. After a separate analysis for the left and right eye in 3C for 1st head-sweep direction FHSD, why do the authors combine all four data categories ($+90^\circ$ LBO, $+90^\circ$ RBO, -90° LBO, and -90° RBO) for the turn direction analysis?
31. Figure 3D. Axis is truncated, I recommend using point markers and dots instead of bars. Showing a delta of turn direction (similar to Figure 1E) might be helpful.
32. Figure 3. After earlier Figures, in which four navigation measures are shown (turn magnitude, turn direction, 1st head-sweep acceptance rate, and 1st head-sweep direction), why are only two of these shown in this Figure? Why did the authors not show data for turn magnitude, and 1st head-sweep acceptance rate?

33. Figure 3E. Why is mean run change introduced here when it was not included in previous Figures? Do larvae not perform run changes in temporal gradients and/or in head-on spatial taxis?
34. On lines 248-250, the authors state that the evidence in Figure 3 shows that turn direction “depends not only on temporal cues (head-sweep acceptance rate) 6, but also on spatial cues (for the first head-sweep direction).” How do the authors know that temporal information is gathered only during head-sweeps, and not during the preceding runs?
35. On lines 248-250, how do the authors relate first head-sweep acceptance rate (FHSAR, a temporal gradient metric) to head-sweep acceptance successes, i.e. head-sweep acceptances in the spatial gradient that result in successful navigation away from the light?
36. Figure 3. Can the authors be sure that the behavioral changes are due to the interaction of the BO ablation with light? Were controls with ablated BO larva done under no light or constant light?
37. In Figure 4, the authors present a third key result: that Rh5 is the crucial photoreceptor for run changes and head sweep biases.
38. Figure 4-6. Some charts have the dependent axes anchored on the right, while other has it anchored on the left. This is confusing for the reader. I recommend all axes are anchored on the left.
39. Figure 4. shibirets is a temperature-sensitive neural inactivator, so a conventional method would be to compare animals before and during heat (within-subjects design) or heated versus unheated animals (between-groups design). This can potentially account for possible genetic differences between strains, along with heat effects and specific thermogenetic effects. However, unheated controls are not reported. Why not?
40. Figure 4A-B. Of the five gradient-relevant variables introduced earlier, only two are shown. Did the authors shop select variables that best supported their case? For example, what is the effect of Rh5 silencing on turning direction? I recommend showing all relevant variables.
41. For the temporal gradient experiment, there is consistency across figures. The reported variables are FHSAR and delta turn angle, and these are reported in every relevant figure. Does mean run change have no relevance in the temporal gradient?
42. For directional light source experiment, there is less consistency across figures. Navigation index is reported in Figure 2, but not Figures 3-6. Similarly turn direction was not reported in Figures 4-6. Mean run changes was omitted in Figure 2. Why were variables not reported in these eight cases?
43. Figure 4A. There is a clear genetic effect of the Rh6-Gal4, which increases run change bias as much as Rh5>shits decreases it, but this is not discussed. While such changes may be common, I recommend that the authors provide some explanation.
44. Figure 4A-B. The authors do not disclose the effect of Rh5 inhibition on turn direction, which would seem to be the most important outcome from a FHSAR bias. If FHSAR bias is impaired, but turn direction is not, this might affect the reader’s interpretation of the importance of this receptor.
45. Figure 4C. For Rh6>shits, the difference between decreasing light intensity and increasing light intensity first head-sweep acceptance rates (FHSAR) is around 5%. For the two relevant controls they are: +/Rh6 = ~16% and +/shits = ~7%. If we take the average of these two control values, the expected difference is ~11%. Does a delta acceptance rate reduction from 11% to 5% justify the statement that “the Rh6-PRs are uniquely essential for a temporally coded navigational strategy”? If we were to take the most ‘pessimistic’ control, then the drop is from 7% to 5%, a functional reduction of just 2%. If we are to accept FHSAR bias as the relevant measure for temporally-coded navigation, then it would indicate that the Rh6-PRs are at best required for slightly more than half of this navigation strategy.
46. Figure 4C. As mentioned above, the axis is truncated at 50%, visually overemphasizing the magnitude of FHSAR bias in all conditions.
47. Figure 4C. In this figure, the FHSAR appears overall higher, with a reduced FHSAR bias (e.g. 79% versus 70% for Rh5/+), when compared to Figure 1F (63% versus 51%). Are these differences a result of the higher temperature (32°C) at which the experiment was conducted? Permissive controls could help account for this..

48. Figure 4D. There is a clear genetic effect of the Rh6-Gal4, which increases delta of mean turn magnitude, but this is not discussed. Also, Rh6>shits is normal by comparison with +/-shits, but this is not discussed. Have the authors have selectively chosen controls that support their case?
49. Figure 4C-D. If the turning magnitude was reported as a delta between the two temporal light conditions, why wasn't the FHSAR?
50. Figure 5. The allOLPs are required for the FHSAR bias, but have no effect on the two gradient navigational variables shown (but see point above about variable choice). Were turn direction and navigation index were calculated, but were not reported? Did allOLP silencing have no effect on these variables?
51. Figure 5 and Lines 331-344. If we assume allOLP silencing had no effect on turn direction and/or navigation index, this raises a question: if temporal processing has no impact on gradient phototaxis, how important is it to real-world navigation?
52. Lines 351-353. "Animals in which PVL09 is genetically silenced are able to perform both spatially encoded phototaxis strategies (first head-sweep direction and steering within runs) (Fig. 6A, B)." This statement is technically correct. However, depending on the control, PVL09>shits larval run change angles are either 2-fold or 4-fold less acute than control animals' run changes. Their FHSD bias is also reduced relative to controls. Both of these reductions are statistically significant for at least one of the two controls, perhaps by 2-fold. How do the authors justify this interpretation of the data when the PVL09 neurons are contributing to these two strategies?
53. Figure 6B. Does the FHSAR phenotype in PVL09>shits larvae translate into a deficit in successful head sweeps, a strategy thought to depend on temporal processing?
54. Figure 6B-C. In 6B, the dependent variable concerns a binary choice, so should be centered at 50%, but the bars extend to the right. In 6C, the variable is not binary, so should not be centered at 50%, but the axis is truncated.
55. Figure 6C-D. Do the processing defects in allLANs>shits larvae (as revealed by Fall four reported metrics) translate to defects in other gradient strategies (turning direction and/or successful head sweeps)?

Reviewers' comments:
Reviewer #1 (Remarks to the Author):

This paper examines negative phototaxis in *Drosophila* larvae. Using two different light presentation paradigms, the authors demonstrate that larvae can use both spatial and temporal cues to bias navigation behavior. Consistent with the idea that larvae use spatial cues for navigation, ablation of either the left or right eye interfered with putative spatial navigational strategies. Finally, the authors used genetic silencing experiments to dissect the role of photoreceptors and LON neurons in these two strategies, and found distinct roles for different neuron types.

Overall I found this paper very interesting. The topic of navigation is of great interest to many researchers, and the dissection of visual navigation into spatial and temporal components is novel. My enthusiasm was lessened somewhat by a difficult exposition. In particular, the methods used to analyze the data are not adequately documented, and insufficient detail is provided in the main text and legends in order for a reader to easily evaluate the material presented in the figures. The paper could also be strengthened by the addition of some controls, and by further experiments looking at the interaction of spatial and temporal navigation modes.

Major concerns:

1) The Methods do not adequately document the criteria for identifying head sweeps, runs, etc. No quantitative criteria are given for these designations, rather a previous paper is referenced. Quantitative criteria must be included, as well as some sense of the degree to which the data could be easily parsed into these different categories.

We apologize for not including this information previously. We have now added all information on how behavioral parameters were identified in the corresponding part of the "Methods section". We adhere with the definitions and thresholds of previous studies (Scantelbury et al. 2007, Gershow et al. 2012, Kane et al. 2013, Humberg and Sprecher 2017). This information is now included under "Navigational parameters" (pages 26- 27, lines 800-818).

2) A shorthand description of how each of the measures shown in the figures was calculated (eg "delta of mean turn magnitude 0° vs 180°") should be provided in the main text and/or figure legend, at least for the first case of each of these appearing. Without this it is very difficult for a reader to evaluate the figures.

This is indeed a good suggestion. We have now included a short definition of how each behavioral parameter is calculated in the main text for the first time of each of these appearing (for data shown in figures) and in figure legends for data shown in supplementary figures (page 7, lines 192-196; page 8 lines 218-220; page 9 lines 251-262).

3) The authors should provide some control for the effects of ablation on navigation and movement. A sham lesion of another group of neurons would greatly strengthen the results in figure 2. The authors should also analyze whether unilaterally lesioned larvae show any differences in the frequency or magnitude of head sweeps.

This is a valid point. We have performed a set of additional experiments to assess the effects of eye ablation. We have also included the required behavioral information in supplementary Figures 2 and 3. We have performed additional sets of experiments running one-eyed larvae in both the temporal and directional assays. For a better, direct comparison we have tested the identical group of animals in both assays as well as constant darkness. These experiments show that the biased behavioral parameters in the spatial assay are not present constant darkness (Fig. 2, 3). Moreover, behavioral parameters that depend on temporal information are still present in one-eyed larvae (Fig. 1D, F). The results show that the magnitude of head-sweeps was unaffected by ablation (sup-Fig 3B). However, one-eyed larvae show an increased turn frequency (sup-Fig 2A).

4) The effect of silencing Rh6 neurons on temporal sensing is not very striking (figure 4C). Can the differences between Rh5 and Rh6-mediated behavior be strengthened by performing temporal or spatial experiments with narrowband light stimuli?

Thank you for proposing this experiment. Based on this suggestion we performed two sets of experiments to further support the role of Rh6 neurons in temporal coding.

First, we used yellow LEDs in the temporal light gradient (sup-Fig. 7). To control for possible effects of the lower intensity we performed experiments in the temporal assay also with a blue/green LED

combination with lower energy-equal light intensity (sup-Fig. 7). According to the published absorption spectrum for Rh6 (Salcedo et al. 1999) only Rh6-PRs should detect the yellow light and mediate behavior in relation to this light.

Second, we used the same LEDs in the directional light gradient. Using yellow light, we do not observe a bias in spatial navigation parameters (steering in runs, first-head sweep direction, turn direction), while these parameters are biased in blue/green light conditions. Moreover, yellow light is sufficient to elicit a significant bias in 1st head-sweep acceptance and turn magnitude, however at a lower rate.

This experiment further supports the effect of Rh6-PRs on temporal sensing, as we could observe behaviors, which depend on temporal light information processing, by stimulating larvae with yellow light.

These experiments are now included in the manuscript on page 13, lines 374-386.

Please see also the answer to the next point.

5) It would be interesting to look at the relationship between temporal and spatial navigation strategies. For example, how is phototaxis impaired when Rh6 or iOLP silencing is combined with unilateral eye ablation? Does this completely abolish phototaxis?

This is indeed an interesting suggestion. We have now performed a set of experiments, in which we silenced Rh6-PRs in one-eyed larvae.

For these experiments, we used the effector shiTS fused to GFP, in combination with laser ablation. We tested these animals with only functional Rh5-PRs in one eye in both the directional and the temporal light source assay (sup-Fig. 8). Interestingly, phototaxis is not completely abolished in one-eyed animals lacking functional Rh6-PRs. While temporal navigational parameters (first head-sweep acceptance rate and turn size) are not biased. This further supports, that Rh6-PRs are involved in this behavioral parameter.

Interestingly, the overall 1st head-sweep direction and turn direction were biased away from the light. These biases are most likely due to the ability of Rh6>shiTS unilateral sensing larvae of biasing the first head-sweep and turn direction when the functional eye is on the body-side facing the light source (sup-Fig. 8). These experiments are now included in the manuscript on page 13, lines 387-393.

6) The distinction between the effects of PVL09 silencing and allLaN silencing (figure 6) seems a bit forced. Overall it appears that these two manipulations had similar effects for all parameters measured, however, the PVL09 controls also had reduced behavior, making the difference not statistically significant. It seems plausible that both of these groups of neurons play similar roles— something that will probably be resolved by physiology.

We completely agree with you and the other reviewer that PVL09 seems to have a central function in phototaxis, as compared to the impact of silencing all-LaNs, while both types of VPNs show overlapping effects. We have modified the manuscript accordingly in the results and discussion.

We modified the results section, which reads now:

“Animals in which PVL09 is genetically silenced are able to perform two spatially encoded phototaxis strategies (first head-sweep direction and steering within runs) (Fig. 6A, B). However, the behavioral biases were strongly decreased in comparison with at least one of the respective controls, suggesting that PVL09 could be involved in mediating both behaviors.” (Page 15, lines 444-449)

The corresponding section in the discussion reads now:

“Overall it appears that also silencing PVL09 had similar effects for all the navigational parameters. It seems plausible that all LaNs and PVL09 possess similar functional roles and contribute to the performance of all measured navigational strategies. Future physiological experiments on these circuit motives will probably help to address further the function of the different interneuron subtypes.” (Page 22, lines 659-664).

7) Previous studies have looked at the effect of temporal changes in light intensity on turn rate and not just percent of first head sweeps accepted. How is turn rate affected by the various manipulations performed in this study?

We agree that the turn rate is a relevant parameter and apologize for not including it previously. In the revised manuscript, we have included the data on turn rate for temporal experiments.

All genotypes and controls show a higher turn frequency in phases of light intensity increase compared with phases of light intensity decrease.

Comparing the experimental genotypes with the respective controls:

- 1. Both PR-subtypes show a decreased turn rate delta compared with their controls, suggesting that both PR-subtypes are required for this navigational parameter.*

2. *Silencing PVL09 and all LaNs (but not solely PDF-LaNs) show also decreased turn rate delta compared with their controls.*
3. *Silencing allOLPs show also decreased turn rate delta compared with their controls. However, silencing pOLP alone shows a decreased turn rate only in comparison with one control.*

The turn rate bias was not lost in any of the tested neurons or group of neurons.

Minor comments:

1) It would be helpful to include a reference to the diagram in figure 5 when describing the anatomy of of the LON in the introduction.

This is a good suggestion. We have added this reference to figure 5A in the introduction.

2) intro line 94: “both navigational strategies are proposed to be mediated by temporal processing” It would be helpful to explain what is meant by temporal processing.

Thank you for this suggestion. We have modified the text describing temporal processing in the introduction in more detail. This section reads now

“Both navigational strategies are proposed to be mediated by temporal processing of light information during head casts⁶. In a light-scape of spatially varying light intensities, a head-sweep will lead to a change in light intensity at the larval eyes over time. Larvae could be able to compare the present with the past light intensity. For example, a detected decrease in light intensity (between these two time-points) during a head-sweep should result rather in accepting than rejecting this head-sweep and as consequence turning towards the darker direction.” (Page 4, lines 94-102)

3) Previous studies have mostly concluded that larvae use temporal processing for navigation and have also found that Rh5 are the main photoreceptors involved. Yet this study shows that Rh5s are mostly involved in spatial processing. This should be addressed in the discussion.

It is indeed an intriguing point that the temporal role of Rh6-PRs was not previously identified. We now included a section in the discussion to highlight this in a wider context. (page 19, lines 572-575).

4) Discussion line 427-430. There is a great deal of evidence for temporal processing in olfactory navigation in adult flies.

Thanks for this comment. We removed the statement about adult chemotaxis.

Reviewer #3 (Remarks to the Author):

1. The authors examine how binocular vision and two photoreceptors mediate the navigational functions of the *Drosophila* larva.
2. The problem that the authors are tackling is very interesting. □
3. Figure 1D and Results lines 161-200. The authors present a key result, that the direction of temporal light changes induces a bias in larval turning, whereby diminishing light increases the acuteness of turning. The data show clearly that larvae are able to detect temporal changes in brightness. □

Thank you for these comments.

4. Figure 1D and Methods lines 624-637. For the light gradient over time, what is the justification to use a 23.5 s gradient? □
5. Figure 1D. How does this rate of brightness change compare to the brightness change that a larva would experience during a head sweep? □
6. Figure 1D. Have the authors estimated (by calculation) the expected brightness change in a head sweep in the gradient setup? □
7. Figure 1D. How does this rate of brightness change compare to the brightness change that a larva would experience during a run? □

The 23.5 seconds interval was previously chosen for practical experimental reasons. We have now performed two additional set of experiments to further assess the impact of light gradient over time. We either used half or double of the time of light increase/decrease (11.75s and 47s) (sup-Fig. 1A, B). In both experimental conditions, we observe a bias in 1st HS acceptance and turn size (sup-Fig. 1C, D). We observe increased values for steeper temporal gradients.

We have further calculated the relative change in light intensity in the spatial assay and included this information in the methods section "visual stimulation".

The slowest temporal intensity ramp would match best to what a larva could sense when it is running straight towards the light. However, in case larvae have directional eyes (our data suggest this) than the light intensity changes might be greater/faster than this. What a larva is sensing during a head-sweep can just be speculated and the signal-to-noise ratio might vary a lot with respect to larval heading direction.

Please note also that changing the step size of the intensity change from 1.5 $\mu\text{W}/\text{cm}^2$ each 100 ms down to 0.72 $\mu\text{W}/\text{cm}^2$ each 100 ms (supplementary Figure 7) results in similar behavioral biases.

We completely agree with you, that our wording was not appropriate, we changed it accordingly. (page 7, lines 185-189)

8. Figure 1E. The axis appears to be mislabelled "0° vs 180°." In the legend, there is mention (lines 807–813) of a yellow and purple compass, but there is no corresponding image in Figure 1. □

We apologize for this. We corrected both mistakes.

9. Figure 1E. A delta is reported. However, the raw observed angle ° values are not. Why were the raw ° values not reported? □

We completely agree with you that the actual raw observed ° value may be more informative to the reader.

We have therefore changed this in all these figures and now display raw observed ° value and not the delta.

10. Figure 1E. Why were the angle ° values in the temporal gradient not directly compared with the angle values in constant light or no light from the experiment in Figure 1C? This would allow the reader to contextualize the angles of turning. □

Thank you for this suggestion. We completely agree with you that this makes it much easier for the reader and is more informative. (Fig. 1D, F; sup-Fig. 2)

11. Figure 1E. The cartoon is somewhat confusing, could be made clearer by having the larva side-by-side rather than back-to-back, and by putting the actual turn angle values next to the larva drawings, perhaps with an angle vertex. □

Very good suggestion. We modified the cartoon, in order to make it clearer.

12. Figure 1F. At first glance, there appears to be a large difference in head-sweep acceptance rate. From the relative lengths of the bars, it seems that larvae are greatly more likely to accept a head sweep if it occurs during decreasing light. However, the axis has been truncated below 50%, so the difference is ~62% versus ~51%, a difference of only 11%. I recommend that either (1) truncated axes are removed from the manuscript or (2) the bars in such cases are replaced by point markers (e.g. circles) instead to indicate the mean. □

13. Figure 1F and all subsequent Figures. The use of bars-only goes against Nature Press' recommended practice to show all individual observed values ["Show the dots in plots." Nature Biomedical Engineering, 1(5), 2017]. □

Thank you for your suggestions for modifying the display of the data. We completely agree with you that this will give much more information and makes it easier for the reader to evaluate the data. We have now modified the figures by displaying all relevant data in the following ways also following "Nature Press" recommendations:

- The axis always starts at 0 (no cropped version)
- We decided to display a bar (overall mean) and errorbars (sem) and adding the values either as numbers or as dots.

For first head-sweep acceptance rate:

The first number is the number of accepted head-sweeps and in brackets is the total number of head-sweeps (accepted + rejected ones). Like this it is much easier for the reader to evaluate the data.

For first head-sweep direction and turn direction probability:

The first number is the number of head-sweeps/turns directed away from the light and in brackets the total number of head-sweeps/turns (away + towards the light source).

For non-binomial data, we added instead of these numbers dots (indicating the mean of the individual experiments), as suggested by you.

This information is now also depicted in figure legend and method section.

14. Figure 1F. The axis orientation is reversed relative to Figure 1E. I can see no reason to do this, and makes it confusing for the reader. Convention would suggest placing zero on the left. □

Thanks for that suggestion. We changed it in all following figures.

15. Figure 1C and 1F. The use of a pie chart in one and a bar chart in the latter inhibits the reader from making a comparison between head-sweep acceptance rates for the four conditions. □

This is indeed a valid point. We adopted the corresponding figure accordingly.

16. For Figure 1F and all subsequent charts showing % values for a binary choice, the authors used the two-tailed exact binomial test. For these data, how was the error bar SEM calculated? □

Thank you for this question. We calculated the SEM as following:

$$\frac{\sqrt{\text{probability} \cdot (100 - \text{probability})}}{\sqrt{\text{total number of events}}}$$

We are now also giving this information in the Methods section "statistical analysis". (Page 30, line 939)

17. Figures 2-3. The authors present a second key result: that larvae use both eyes to navigate with respect to

light. Having both eyes intact is relied upon by several navigational motor functions: head sweep bias, turning direction, and run changes. □

18. Figure 2 shows convincingly that larvae favor turns in the direction of an ablated BO, by about 15%, and that this largely derives from a first head-sweep direction in favor (by about 12%) of the ablated eye. □

19. Figure 2B. The axis is unlabelled. From the legend, it should be "Navigation Index." □

We are sorry for that. We added the label.

20. Figure 2B. The Navigation Index has no negative control. For example, what is the Navigation index for larvae when the directional source is turned off, i.e. under red light-only conditions? □

We have now included the navigation index of one-eyed larva for no-light condition as control.

21. Figure 2B. More generally, no locomotion data are shown. For the BO-ablated animals, is the locomotion (turning metrics, speed) normal? □

We now include these behavioral parameters for BO ablation experiments in all the relevant figures as well as in supplemental Fig 2 and 3. Ablation has no effect per se on larval speed or head-sweep size (sup-Fig. 3).

22. Figure 2C. Truncated axis, but since it is centered around 50%, I recommend replacing bars with mean point markers and dots. □

23. Figure 2D. Axis truncation which should be fixed, preferably by extending to zero. □

24. Figure 2E. As per 2C. □

We have changed this accordingly (see also comment 12 above).

25. Figure 2. The diagram of the ablated BO is drawn as though it is on the left eye. However these data are labelled "towards the ablated eye," suggesting that data from both left- and right-ablated animals was used, and combined into an achiral data set. If this is correct, then the authors need to mention this somewhere. It is of importance since the control data are chiral, as are the data in Figure 3C. □

This is indeed true. We have therefore modified the text that in this visualization of the experiments we have used eye-ablation data for both eyes in an achiral data set. These sections now read as follows:

"When initially heading directly towards a light source, larvae should show no preference to turn left or right, as either direction is away from the light, and control larvae are equally likely to turn left or right⁶ (Fig. 2A). For eye-ablation experiments we merged the data in an achiral data-set. Larvae with either the left or right eye ablated bias their turn direction to the side of their ablated eye suggesting a spatial comparison of light input between the left and right eye¹ (Fig. 2A)." (Pages 9-10, lines 263-269)

"We next analyzed if larval phototaxis benefits from biasing the first head-sweep direction when larvae are heading perpendicular to the light source. When heading towards +90° larvae lacking the left eye possess the functional eye on the side where the light source is positioned, while if these animals head towards -90° their functional eye is oriented away from the light source (Fig. 3A). The same is true in an inverted fashion for larvae lacking the right eye (Fig. 3A). Therefore, we have used again eye-ablation data for both eyes in an achiral data-set. If larvae use a spatial integration mechanism to avoid a light source, they might bias the first head-sweep direction away from the light source and/or steer away from it within runs, when heading perpendicular to the stimulus⁶ (Fig. 3B). Thus, we grouped all events when animals had the remaining eye either on the body side towards the light-source or away from it (Fig. 3A)." (Page 10, lines 283-295)

26. In Figure 3, the authors show that: (1) the first head-sweep direction (FHSD) bias is impaired when an ablated BO faces towards the light; (2) that BO ablation reduces turn direction bias; (3) that run changes become more obtuse when the ablated BO is facing the light source; and (4) light-avoiding maneuvers are close to normal when the intact BO faces the light. □

-no comment needed-

27. On lines 263–265, the authors state “That larvae are able to bias their movements away from the light when their functional eye is on the light-exposed side is suggestive for directional eyes.” Two models for binocular function in this case could be (1) an intensity comparator or (2) directional eyes. The intensity comparator should be incapacitated with removal of one eye, since no comparison can be made. A single directional eye should have 100% function in its relevant direction. As the larvae have close to normal FHSD and run change acuity functions on the intact BO side, the authors’ data would support the latter model for +/-90° navigation. □

Thank you for this comment. We have added a sentence in the results sections.

The section reads now:

“Binocularity could either be necessary for the function of an intensity comparator or directional eyes. Loss of one eye should lead to a completely disabled intensity comparator. That larvae are able to bias their movements away from the light when their functional eye is on the light-exposed side is therefore suggestive for directional eyes.” (Page 11, lines 323-328)

28. Figure 3C. Larvae 1st head-sweep direction (FHSD) bias is unaffected when the ablated eye is distal to the light source, but is eliminated when the ablated eye is proximal to the light source. As 50% is the value of indifference, I recommend replacing bars with point markers. □

Thank you for this suggestion, we have modified this accordingly (see also comment 12 above).

29. Figure 3C. Do the authors have reason to believe that Drosophila larvae have a handedness in their light avoidance? Otherwise, why is it necessary to analyze the head-sweep direction independently for left-ablated and right-ablated BO? Would it make sense to make an achiral analysis (as appears to have been done for Figure 2) with respect to the light-proximal and light-distal ablated BO instead? □

We agree with this suggestion. We combined the data as proposed.

In order to exclude an effect of handedness in wildtype animals we have included both +90° and -90°.

30. Figure 3D. After a separate analysis for the left and right eye in 3C for 1st head-sweep direction FHSD, why do the authors combine all four data categories (+90° LBO, +90° RBO, -90° LBO, and -90° RBO) for the turn direction analysis? □

Thanks for this suggestion. We combined the data as proposed (see comment 29).

31. Figure 3D. Axis is truncated, I recommend using point markers and dots instead of bars. Showing a delta of turn direction (similar to Figure 1E) might be helpful. □

Please see our answer to point 12.

32. Figure 3. After earlier Figures, in which four navigation measures are shown (turn magnitude, turn direction, 1st head-sweep acceptance rate, and 1st head-sweep direction), why are only two of these shown in this Figure? Why did the authors not show data for turn magnitude, and 1st head-sweep acceptance rate? □

This is a valid question. We apologize for not clearly depicting this subject in the previous version of the manuscript. We now coherently use the two complementary assays (spatial versus temporal gradients) for their respective navigational parameters.

We do not want to make a statement, if a spatial or temporal cues (or the behaviors based on these cues) are more relevant or impacting for navigation under real world conditions. In laboratory conditions the use of both assays allow the controlled assessment of spatial or temporally based strategies.

We have now changed the manuscript and added two section reading:

“In a first assay we exposed animals to temporally varying, but spatially uniform light-scape to test uniquely for temporal changes in light stimulation” page 7, lines 187-189

“This set-up creates a spatial light gradient, which can be used by larvae to perform navigational strategies based on left-right comparisons, however it may not exclude that a larva is detecting temporal changes in light intensity while moving.” (Page 9, lines 248-251) line:

33. Figure 3E. Why is mean run change introduced here when it was not included in previous Figures? Do larvae not perform run changes in temporal gradients and/or in head-on spatial taxis? □

Larvae indeed do not bias their steering within runs in (°/cm) in relation to temporal intensity changes. We have now included this information in supplemental figure 2B. Moreover, we also included this navigational parameter in head-on spatial taxis in Fig. 2.

34. On lines 248-250, the authors state that the evidence in Figure 3 shows that turn direction “depends not only on temporal cues (head-sweep acceptance rate) θ , but also on spatial cues (for the first head-sweep direction).” How do the authors know that temporal information is gathered only during head-sweeps, and not during the preceding runs? □

We fully agree that this cannot be completely excluded. Therefore, we deleted this sentence.

35. On lines 248-250, how do the authors relate first head-sweep acceptance rate (FHSAR, a temporal gradient metric) to head-sweep acceptance successes, i.e. head-sweep acceptances in the spatial gradient that result in successful navigation away from the light? □

We apologize for the overstated wording of this section. Since a direct comparison of the two assays is not possible we have deleted this sentence. (see also above point 34)

36. Figure 3. Can the authors be sure that the behavioral changes are due to the interaction of the BO ablation with light? Were controls with ablated BO larva done under no light or constant light? □

Thank you for raising this point. We have now performed a set of experiments to address this question. We have repeated the ablation experiments and performed appropriate control experiments with the same animals. The unilateral seeing animals were now navigate not only in a light gradient, but also in the temporal assay and in absence of any stimulus (Fig. 1, 2, 3; sup-Fig. 2., 3.; 4). All these results exclude that eye ablation had an effect per se on larval navigation, but that in fact it has an effect on some locomotion parameters. The observed behavioral changes in the light gradient are indeed based on the interaction of the BO ablation with light.

37. In Figure 4, the authors present a third key result: that Rh5 is the crucial photoreceptor for run changes and head sweep biases. □

38. Figure 4-6. Some charts have the dependent axes anchored on the right, while other has it anchored on the left. This is confusing for the reader. I recommend all axes are anchored on the left. □

Indeed, we fully agree with this and have changed figures accordingly.

39. Figure 4. shibirets is a temperature-sensitive neural inactivator, so a conventional method would be to compare animals before and during heat (within-subjects design) or heated versus unheated animals (between-groups design). This can potentially account for possible genetic differences between strains, along with heat effects and specific thermogenetic effects. However, unheated controls are not reported. Why not? □

Thank you for pointing this out. It is indeed true that a cold-control is often used as control in behavioral assays. However, for larval navigation an increased temperature changes behavior severely: increasing larval crawling (increasing run speed and length) or decreasing the turn rate (decreasing the number of first head-sweeps) (Ohyama et al. 2013, Pauls et al. 2014). Since this directly impacts parameters of navigation strategies we use the appropriate genetic control at high-temperature.

We apologize for not giving a statement to this issue in the first version of the manuscript. However, we added now a statement in the text:

“To overcome potential developmental defects, we genetically silenced either Rh5- or Rh6-PRs by expressing a temperature sensitive dominant negative form of Dynamin (UAS-shi^{TS}), specifically in either PR-type. Since an increased experimental temperature generally changes larval behavior (Supplementary Fig. 5) we here compare cross with its proper genetic control at 32°C (Ohyama et al. 2013, Pauls et al. 2014).” (Page 12, lines 341-346)

To clarify the effect of high temperature of wildtype larvae we have further, we added data of wild-type larvae performing in both behavioral assay at temperature of 32°C (Supplementary Fig. 5).

40. Figure 4A-B. Of the five gradient-relevant variables introduced earlier, only two are shown. Did the authors shop select variables that best supported their case? For example, what is the effect of Rh5 silencing on turning direction? I recommend showing all relevant variables.□

Thank you for pointing this out. We have now included the turn direction in Fig. 4-6 and navigation index in supplement fig 6, 9 and 10. Rh5-PRs are indeed necessary for biasing the turn direction.

41. For the temporal gradient experiment, there is consistency across figures. The reported variables are FHSAR and delta turn angle, and these are reported in every relevant figure. Does mean run change have no relevance in the temporal gradient?□

We added a supplementary figure (2B), which shows that indeed steering within runs is not effected by light in- or decrease.

42. For directional light source experiment, there is less consistency across figures. Navigation index is reported in Figure 2, but not Figures 3-6. Similarly turn direction was not reported in Figures 4-6. Mean run changes was omitted in Figure 2. Why were variables not reported in these eight cases?□

We apologize for the inconsistency; it is indeed true that we previously primarily displayed the respective navigation relevant parameters. We adapted all figures accordingly and now consistently show all navigational parameters in the corresponding figures.

Please note, that the figure 2 and 3 belong to the same experimental data set, but is analyzed separately accordingly to the heading direction of the larvae.

43. Figure 4A. There is a clear genetic effect of the Rh6-Gal4, which increases run change bias as much as Rh5>shits decreases it, but this is not discussed. While such changes may be common, I recommend that the authors provide some explanation.□

This is indeed a valid point. We apologies for omitting a comment on the behavioral difference of this control. We have now included a sentence depicting this effect of run change bias in the result section page 13, line 369-372.

The sentence reads as follows:

“Since silencing Rh6-PRs shows a significant different mean run change and turn magnitude to the Rh6-driver line control, but not to the UAS-shits control we cannot exclude a behavioral contribution of Rh6-PRs in these strategies.”

44. Figure 4A-B. The authors do not disclose the effect of Rh5 inhibition on turn direction, which would seem to be the most important outcome from a FHSD bias. If FHSD bias is impaired, but turn direction is not, this might affect the reader's interpretation of the importance of this receptor.□

Thanks for this suggestion. Rh5-PRs are indeed necessary for biasing the turn direction (Fig 4C). We are now giving the data for turn direction in all relevant figures.

45. Figure 4C. For Rh6>shits, the difference between decreasing light intensity and increasing light intensity first head-sweep acceptance rates (FHSAR) is around 5%. For the two relevant controls they are: +/Rh6 = ~16% and +/shits = ~7%. If we take the average of these two control values, the expected difference is ~11%. Does a delta acceptance rate reduction from 11% to 5% justify the statement that “the Rh6-PRs are uniquely essential for a temporally coded navigational strategy”? If we were to take the most ‘pessimistic’ control, then the drop is from 7% to 5%, a functional reduction of just 2%. If we are to accept FHSAR bias as the relevant measure for temporally-coded navigation, then it would indicate that the Rh6-PRs are at best required for slightly more than half of this navigation strategy.□

We agree with this comment and have changed the statement accordingly. The sentence reads now “In summary, the Rh6-PRs are required for a temporally coded navigational strategy, while the Rh5-PRs are solely essential for spatially coded navigational strategies.” (Page 13, lines 394-397)

Please note that we have included additional support that functional Rh5-PRs are not necessary for mediating this behavioral parameter (Fig. 4D). Only silencing Rh6-PRs results in first head-sweep acceptance rates which are statistically not different from each other.

Further, we have performed different additional sets of experiments to support the role of Rh6-PRs in the first head-sweep acceptance rates (see also comment Reviewer 1 comment 4 and 5).

First, we used yellow LEDs as light source for the temporal and directional assay (sup-Fig. 7). According to the calculated Rhodopsins absorption spectra, only Rh6-PRs should detect yellow light and therefore being able to mediate behavior, accordingly (Salcedo 1999).

In the temporal assay, yellow LEDs were eliciting a first head-sweep acceptance rate and a turn size bias (sup-Fig. 7F, G), however at lower rates.

In the spatial assay, yellow LEDs were not eliciting a single parameter, whereas blue and green LEDs of the same intensity did (sup-Fig. 7A-D).

Both experiment further supports the effect of Rh6-PRs on temporal sensing, as we could observe behaviors, which depend on temporal light information processing, by stimulating larvae with yellow light.

Second, we combined silencing of Rh6-PRs with eye-ablation.

Although in these experiments were the first head-sweep acceptance rates unbiased (sup-Fig. 8E).

All these additional experiments further support that Rh6-PRs are required for a temporally coded navigational strategy – namely first head-sweep acceptance rate.

46. Figure 4C. As mentioned above, the axis is truncated at 50%, visually overemphasizing the magnitude of FHSAR bias in all conditions. □

We adopted all figures for this (see also comment 12 above).

47. Figure 4C. In this figure, the FHSAR appears overall higher, with a reduced FHSAR bias (e.g. 79% versus 70% for Rh5/+), when compared to Figure 1F (63% versus 51%). Are these differences a result of the higher temperature (32°C) at which the experiment was conducted? Permissive controls could help account for this. □

Thank you very much for this question.

Indeed, is temperature modulating this parameter (and others), to account for this we added a supplementary figure showing the behavior of wildtype control animals in both assay at 32°C (sup-Fig. 5.). (see also comment 39)

48. Figure 4D. There is a clear genetic effect of the Rh6-Gal4, which increases delta of mean turn magnitude, but this is not discussed. Also, Rh6>shits is normal by comparison with +/shits, but this is not discussed. Have the authors have selectively chosen controls that support their case? □

We conclusively used the proper genetic control for all experiments. As discussed above in point 43 we have now included a statement on the difference of the Rh6-Gal4 control (page 13, line 369-372)

49. Figure 4C-D. If the turning magnitude was reported as a delta between the two temporal light conditions, why wasn't the FHSAR? □

Thanks to your suggestion from above, we changed all figures and are now only reporting raw values.

50. Figure 5. The allOLPs are required for the FHSAR bias, but have no effect on the two gradient navigational variables shown (but see point above about variable choice). Were turn direction and navigation index were calculated, but were not reported? Did allOLP silencing have no effect on these variables?

We added the data for the turn direction and the navigation index to all relevant figures.

pOLP and POG have not a decreased navigation index and no defect in turn direction bias (Fig. 5D; sup-Fig. 5A).

51. Figure 5 and Lines 331-344. If we assume allOLP silencing had no effect on turn direction and/or navigation index, this raises a question: if temporal processing has no impact on gradient phototaxis, how important is it to real-world navigation? □

Thank you very much for raising this point. indeed, functional OLPs appear not to be necessary for any parameter of the directional light source assay.

However, outdoor experiments (probably a little bit closer to real world conditions) suggest that temporal processing could be important under certain conditions!

We previously reported that in outdoor experiments Rh5- and Rh6-PRs are either necessary or sufficient for light avoidance under cloudy or sunny conditions, respectively (Humberg and Sprecher 2017). The results from outdoors are quite different to what we and others observe under laboratory conditions. Also others (Hinnemann et al. 2010) suggested that larvae could use spatial and/or temporal cues for light avoidance in dependence of the environmental conditions.

We are currently restricted to indoor use of our tracking system, but we aim in the future to develop a system that will allow to perform similar experiments with outdoor natural lighting conditions.

We made statements about this in the text:

“Outdoor experiments combined with the use of our tracking system will be of great interest to further address the individual function of the two PR-subtypes under distinct lighting conditions.”(Page 19, lines 587-589)

“Thus, it is more likely that the use of Rh6 instead of a different Rhodopsin is not essential for the function of this PR-subtype in general, but that it may rather be relevant as adaption to the lighting conditions under which the task of these PR-subtype is most essential.” (Page 20, lines 616-619)

-Please see also our answer to point 32-

52. Lines 351-353. “Animals in which PVL09 is genetically silenced are able to perform both spatially encoded phototaxis strategies (first head-sweep direction and steering within runs) (Fig. 6A, B).” This statement is technically correct. However, depending on the control, PVL09>shits larval run change angles are either 2-fold or 4-fold less acute than control animals’ run changes. Their FHS bias is also reduced relative to controls. Both of these reductions are statistically significant for at least one of the two controls, perhaps by 2-fold. How do the authors justify this interpretation of the data when the PVL09 neurons are contributing to these two strategies?□

Thank you very much for this comment. We agree completely with you and the other reviewer that PVL09 seems to have an important function in phototaxis.

We added additional sentences in the respective results section, which reads now:

“Animals in which PVL09 is genetically silenced are able to perform two spatially encoded phototaxis strategies (first head-sweep direction and steering within runs) (Fig. 6A, B). However, the behavioral biases were strongly decreased in comparison with at least one of the respective controls, suggesting that PVL09 could be involved in mediating both behaviors. Further, these animals possess a decreased turn direction bias in comparison with both controls (Fig. 6C).” (Page 15, lines 444-450)

In the revised manuscript, we added also to the discussion that beside lack of significant statistical differences PVL09 and LaNs might play a role in all navigational strategies. Furthermore, supported by the fact that silencing the LaNs do not result in a completely abolished behavioral bias for most of the behavioral strategies. Meaning that there might be atleast another interneuronsubtyp (most likely pvl09) involved in mediating all the navigational strategies.

This reads:

“Overall it appears that also silencing PVL09 had similar effects for all the navigational parameters. It seems plausible that all LaNs and PVL09 possess similar functional roles and contribute to the performance of all measured navigational strategies. Future physiological experiments on these circuit motives will probably help to address further the function of the different interneuron subtypes.” (Page 22, lines 659-664)

53. Figure 6B. Does the FHSAR phenotype in PVL09>shits larvae translate into a deficit in successful head sweeps, a strategy thought to depend on temporal processing?□

We indeed observe deficits in the head sweep success in animals with silenced PVL09, we describe a significantly lower turn direction bias compared to the respective control values which we have now included in Fig. 6C.

54. Figure 6B-C. In 6B, the dependent variable concerns a binary choice, so should be centered at 50%, but the bars extend to the right. In 6C, the variable is not binary, so should not be centered at 50%, but the axis is truncated.□

Thanks, we changed this accordingly.

55. Figure 6C-D. Do the processing defects in allLANs>shits larvae (as revealed by all four reported metrics) translate to defects in other gradient strategies (turning direction and/or successful head sweeps)?□

We apologize for not giving all the relevant data in the first version. Indeed, allLaNs>shiTS larvae cannot bias their turn direction away from the light source (Fig 6C). Further, the navigation index of these larvae is decreased in comparison to the navigation indices of the respective controls (sup-Fig. 10A).

REVIEWERS' COMMENTS:

Reviewer #1 (Remarks to the Author):

The authors have address all of my major concerns. The supplementary experiments provide a fuller picture of the contributions of the two eyes and of different neuron classes to larval phototaxis. I think the manuscript is important and should be published in Nature Communications.

A few minors comments:

- 1) The description of the larval optic neuropil anatomy in the introduction is still hard for an outsider to read, as it assumes the reader is familiar with the pdf neurons and OLPs. The authors could revise lines 116-140 to lay out the anatomy in a more straightforward fashion. There is also currently a typo in line 121 ("functioning").
- 2) The last paragraph of the introduction is also less clear than it could be. Perhaps the authors could summarize a few key takeaways from their study, instead of listing all their findings, as they currently do here.
- 3) line458-460. This sentence is not grammatical.
- 4) It would be nice to include confocal stacks of the lines used showing which neurons they label, so readers can judge how clean the lines are.

Reviewer #4 (Remarks to the Author):

Second Round Review Humberg et al., 'Dedicated photoreceptor pathways...'

This paper has previously been seen by two reviewers. Both reviewers commended the topic and approach, but had a series of concerns.

In this revision, the authors provide a detailed and thoughtful rebuttal of both reviewers' critiques. Both reviewers shared questions about unclear methods, which are addressed by the authors. Reviewer 1 had methodological questions and requested some control experiments, which the authors performed. The reviewer also suggested an experiment with 'narrowband stimuli'. The authors did the experiments and found supportive evidence for temporal sensing. Lastly, the authors performed a third set of phototaxis experiments in one-eyed larvae, which revealed still some functional phototaxis. All revision experiments are well integrated and discussed.

Reviewer 3 is not negative, finds the problem interesting, but has a long list of very detailed questions, which I find altogether adequately addressed by the authors. The list starts with a question about the odd time gradient value of 23.5 sec, which was due to technical reasons. Importantly, the authors provide other increase increase/decrease times as controls for the revised manuscript. A long list of concerns follows (up to point 35), which were throughout recognized by the authors as justified and see to have been corrected. Point number 36 is a concern about Bolwig organ ablation experiments, which the authors address by control experiments showing that eye ablation per se did not cause behavioral changes. All further points (until number 55!) were again adequately addressed by the authors.

My own reading of the paper is positive. I find the study very through and detailed, technically sound and conceptually novel. I support publication in Nature Communications.

REVIEWERS' COMMENTS:

Reviewer #1 (Remarks to the Author):

The authors have address all of my major concerns. The supplementary experiments provide a fuller picture of the contributions of the two eyes and of different neuron classes to larval phototaxis. I think the manuscript is important and should be published in Nature Communications.

A few minors comments:

1) The description of the larval optic neuropil anatomy in the introduction is still hard for an outsider to read, as it assumes the reader is familiar with the pdf neurons and OLPs. The authors could revise lines 116-140 to lay out the anatomy in a more straightforward fashion. There is also currently a typo in line 121 ("functioning").

We have revised the introduction according to these suggestions. The sections on the anatomy (now line 137-162) has been streamlined in order to eliminate non-essential information.

We apologize for the typo and have accordingly change the sentence of former line 121 (now line 152).

2) The last paragraph of the introduction is also less clear than it could be. Perhaps the authors could summarize a few key takeaways from their study, instead of listing all their findings, as they currently do here.

We have revised the introduction according to the suggestions including the last paragraph. We now highlight most relevant results.

3) line458-460. This sentence is not grammatical.

Thank you for pointing this out. We corrected this sentence. It reads now:
“Further, animals with genetically silenced PVL09 show a reduced performance of the other behavioral strategies including the overall navigation index and the turn rate bias (Fig. 6; Supplementary Figure 10).”

4) It would be nice to include confocal stacks of the lines used showing which neurons they label, so readers can judge how clean the lines are.

Thank you for this comment. In fact, we now not included the anatomical expression of the Gal4 lines, since we have just published precisely this piece of information in great detail in a paper in eLIFE half a year ago (*Larderet, I. et al. Organization Of The Drosophila Larval Visual Circuit. eLIFE 6, doi:10.7554/eLife.28387 (2017)*). We therefore feel that it would be too repetitive to include the same information again and we refer to our recent publication in the manuscript. However, if you feel that this is a critical piece of information we would be happy to perform these experiments and include them here.

Reviewer #4 (Remarks to the Author):

Second Round Review Humberg et al., 'Dedicated photoreceptor pathways...'

This paper has previously been seen by two reviewers. Both reviewers commended the topic and approach, but had a series of concerns.

In this revision, the authors provide a detailed and thoughtful rebuttal of both reviewers' critiques. Both reviewers shared questions about unclear methods, which are addressed by the authors.

Reviewer 1 had methodological questions and requested some control experiments, which the authors performed. The reviewer also suggested an experiment with 'narrowband stimuli'. The authors did the experiments and found supportive evidence for temporal sensing. Lastly, the authors performed a third set of phototaxis experiments in one-eyed larvae, which revealed still some functional phototaxis. All revision experiments are well integrated and discussed.

Reviewer 3 is not negative, finds the problem interesting, but has a long list of very detailed questions, which I find altogether adequately addressed by the authors. The list starts with a question about the odd time gradient value of 23.5 sec, which was due to technical reasons. Importantly, the authors provide other increase/increase/decrease times as controls for the revised manuscript. A long list of concerns follows (up to point 35), which were throughout recognized by the authors as justified and seen to have been corrected. Point number 36 is a concern about Bolwig organ ablation experiments, which the authors address by control experiments showing that eye ablation per se did not cause behavioral changes. All further points (until number 55!) were again adequately addressed by the authors. My own reading of the paper is positive. I find the study very thorough and detailed, technically sound and conceptually novel. I support publication in Nature Communications.

We thank this reviewer for the positive comments on our manuscript.